# Assessing agricultural effects on benthic invertebrate communities in ponds and ditches using δ¹⁵N and δ¹³C isotope niches

Fee Nanett Trau 🔾*, Kathrin Fisch🔾, Stefan Lorenz🔾

Institute for ecological chemistry, plant analysis and stored product protection, Julius Kühn-Institute, Berlin, Germany

* fee-nanett.trau@julius-kuehn.de

## Abstract

Agriculture is one of the dominant types of land use worldwide. The land conversion and intensification of agriculture have severe impacts on biodiversity and the associated ecosystem functions in the production areas and in natural or semi-natural habitats embedded in the agricultural landscape. Ponds and ditches, while vulnerable to the negative impacts of agriculture, are valuable habitats for benthic invertebrates due to their heterogeneity and location in a rather homogeneous landscape. Due to a lack of sensitivity biodiversity metrics based on taxonomy, such as overall taxa number, Shannon or evenness index, might fail to detect changes in communities, which lead to changes in ecosystem functions resulting from agricultural practices. In contrast, functional approaches, such as the use of functional feeding groups and isotopic composition, better describe the utilization of resources by invertebrates and improve our understanding of the impact of agricultural stressors on benthic invertebrate communities in ponds and ditches. Benthic invertebrates, from six different functional feeding groups (FFGs), and water samples were collected from three ponds and four ditches in an agricultural landscape in Brandenburg, Germany. We analyzed the carbon and nitrogen stable isotope ratios of the invertebrates and the pesticides and nutrient residues of water samples. Estimates of community metrics were derived using a Bayesian approach for the two water body types and the FFGs present. A distance-based Redundancy Analysis was carried out to detect the environmental variables that show the maximum correlation with the derived metrics. Benthic invertebrates in ponds occupied a larger total area and showed greater spacing of individuals in the isotope space than communities in ditches, indicating that more resources and/or habitats were available and utilized in ponds. No clear pattern emerged when comparing the FFGs in the two water body types, but some differences were found, for example in FFG *predator*, which were found in the ditches with several different taxa and were thus more diverse. The strong fluctuations in biotic and abiotic parameters led to pronounced differences between the water bodies.

**Data availability statement:** The raw data on the stable isotope composition (δ13C and δ15N) of the consumers are available at the PANGEA repository. Reference to the data set is: Trau, Fee Nanett; Lorenz, Stefan (2025): Pond/ditch benthic invertebrate data on d15N/d13C (northeast-Germany, 2020). PANGEA. DOI: 10.1594/PANGAEA.985189.

**Funding:** This work was funded by the German Federal Ministryof Food and Agriculture (BMEL) via the Agency for Renewable Resources (FNR; project number22012018) based on a decision of the Parliament of the Federal Republic of Germany. The funders had no role in study design, data collection and analysis, decision to publish, or preparation of the manuscript.

**Competing interests:** The authors have declared that no competing interests exist.

Our study showed that benthic invertebrate communities in small unstable systems respond in complex ways to stressors. However, larger datasets may yield in more pronounced patterns on effects of agriculture on stable isotope composition in ponds and ditches.

## 1. Introduction

Agricultural land covers about 38% of the world's terrestrial area, which makes it the predominant land use type [1]. The conversion of forests, wetlands, and natural grasslands to agricultural land has severe impacts on the provision of habitats, biodiversity, and associated ecosystem functions [1,2]. Likewise, the increasing specialization of farmers, the expansion and intensification of agriculture (e.g., by fertilization and pesticide application) further increased homogenization of the landscape, resulting in accelerated habitat fragmentation and biodiversity loss [3]. Those developments continue to exert ongoing pressure on the remaining semi-natural (e.g., hedges, ponds, field margins, ditches) or natural habitats within the agricultural landscape and on the surrounding aquatic and terrestrial environments [4].

Ponds and ditches serve as biodiversity hotspots and act as aquatic corridors within agricultural landscapes, highlighting their importance for conservation efforts [5,6]. Although these small water bodies are usually not included in monitoring programs [5,7] their importance for benthic macroinvertebrate biodiversity and ecosystem services is increasingly gaining recognition in research activities [5,8–13]. This study will contribute to gain knowledge on measuring changes in communities, which is indispensable in order to conserve and promote these aquatic habitats and their biodiversity. Thorough knowledge on the status and indicators for measuring biodiversity in ponds and ditches affected by agricultural activities is needed.

Results from a previous study on ponds have shown that conventional structural biodiversity metrics (e.g., taxonomic diversity, Shannon index, and the EPT index, to a certain degree) do not perform well in detecting stressed communities in agricultural areas due to long-term and ubiquitous persistence of stressors and short stressor gradients [11]. Additionally, species occurrences may be subject to some degree of stochasticity [14] and functional responses can reflect the ecological conditions and habitat change better than taxonomic approaches [15,16]. Therefore, functional indicators and approaches could be promising tools to detect community disturbance and to examine possible direct as well as indirect effects of stressors on benthic invertebrate communities and their functions [11,17,18].

Common traits used to assess the functional diversity of benthic invertebrates are feeding mode, locomotion, maximum body size, and respiration mode [19]. As benthic invertebrates play an important role in the cycling of nutrients and decomposition of organic matter, the analysis of trophic relationships and functional feeding groups (FFGs) are central to understand ecosystem processes [20,21]. Such analyzes also provide valuable insights into the impacts of agricultural activity, including habitat fragmentation, degradation, and pollution, which alter resource quality, quantity, and

type, ultimately influencing community structure [ 17,22–24]. Understanding how species interact with their environment and with each other is therefore essential for interpreting changes in community structure. In this context, the niche concept provides a framework for linking species traits and resource use with ecosystems.

The niche concept is fundamental to understand how communities are organized and how ecological interactions shape their strategies and ecological shifts [25]. The niche of a species describes the range of conditions necessary for the species to exist and reproduce, including its abiotic and biotic interactions within the environment [26]. Niches can overlap when species are co-occurring, which leads to interspecific competition. Higher species richness can lead to greater niche overlaps under environmental disturbance, as for example found in carabid communities [27]. The basic principle of "you are what you eat (plus a few ‰)" [28] has led to the use of stable isotopes to analyze isotopic niches, food web structures, and species interactions in ecosystems [29]. Two elements commonly used for this purpose are nitrogen and carbon. The ratio of $^{15}N$ to $^{14}N$ progressively increases at each trophic level whereas the ratio of $^{13}C$ to $^{12}C$ remains almost unchanged when passing through the food web, but discriminates food sources due to different photosynthetic pathways [28]. Accordingly, $\delta^{15}N$ values are used to determine the trophic position of an organism while $\delta^{13}C$ values provide information on the sources utilized by the organism. Stable isotope ratios of carbon and nitrogen allow the quantification of the isotopic niche of a species or taxonomic group (e.g., Coleoptera, Odonata etc.). From these data, metrics can be derived that reflect isotopic niche width or proxies for resource use, enabling statistical comparisons across broad spatial scales and among communities composed of different species [18,30]. The isotopic niche can thus serve as a proxy for the realized ecological niche of a species (portion of the ecological niche that a species occupies in nature, after accounting for competition and predation [31]) reflecting their use of resources/diet and habitat [32]. The realized niches are linked to land use and agricultural stressors, which alter resource availability and habitat quality. In this study, the term niche will be used in the sense of the isotopic niche as described above.

We aimed to assess the influence of agricultural stressors, measured as water toxicity from pesticide residues and eutrophication, on the isotopic niche of FFGs in pond and ditch communities, using a geometric approach originally developed by [24]. Given that ponds and ditches are likely affected differently by these stressors, with consequent impacts on habitat availability and quality, we hypothesize that such variations ultimately shape benthic invertebrate food webs and niche size [33,34]. Ditches are expected to be more strongly influenced by agricultural stressors than ponds due to their morphological characteristics, which are a rather homogeneous habitat in and around the water body, a directly farmed environment with the location in the field (grassland) without a riparian buffer. In addition, ditches are probably more contaminated, but as they are part of a flow system, pollutants can also be transported downstream. Ponds, on the other hand, are better protected by a riparian buffer from pesticide inputs, but can act as a sink (e.g., sediment deposition and retention due to long retention times and closed systems, biological uptake in plants) and source (e.g., remobilisation from sediment due to disturbances, contaminated groundwater connection) for pesticide residues in the sediment [35,36]. Ponds and ditches representing water body types with varying exposure to agricultural effects (nutrient enrichment, pesticides, and the proportion of surrounding crops) were used to test our hypotheses. We asked the following questions: Can the isotopic niches depict the effects of agriculture on benthic invertebrate communities in the selected water bodies in the study region? Do the observed patterns of isotopic niches of FFGs differ between ponds and ditches? We hypothesize that agricultural activity and low habitat variability in ditches have led to simplified trophic structures, which will be shown by wider isotopic niches, reduced niche differentiation, and larger niche overlap due to reduced differentiation of the FFGs and more generalists present. On the other hand, we hypothesize that trophic structures in ponds are more differentiated and that we observe more specialized groups with less niche overlap.

## 2. Methods and data

### 2.1. Study area

Sampling was carried out in Brandenburg in the "Havelländisches Luch" region (Fig 1). The wet lowlands in the region were drained and an intensively maintained ditch system was created for industrial agricultural production during the

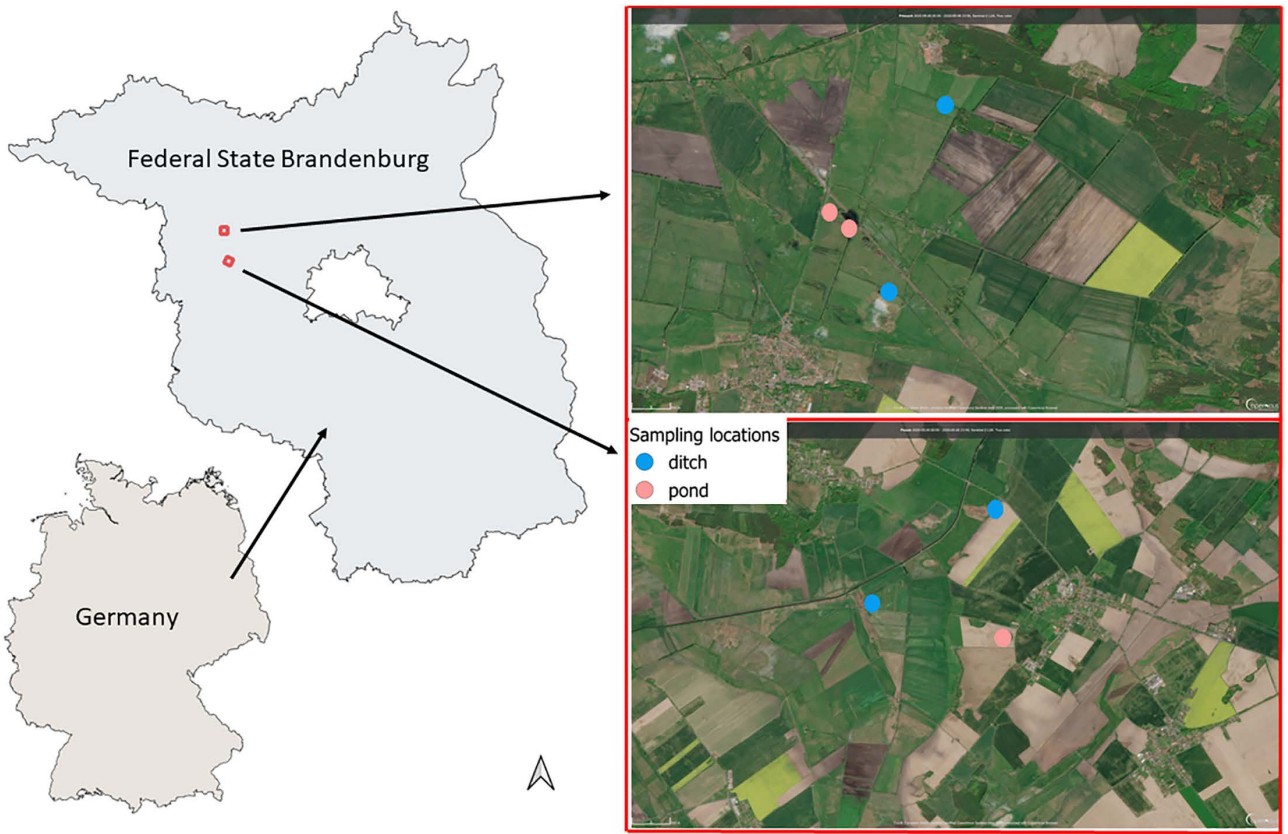

**Fig 1. Map of sampling sites, ditches in blue and ponds in red.** Map data: European Union contains modified Copernicus Sentinel data 2025 processed with Copernicus Browser, Sentinel 2-L2A.

1980s [37]. Traditional extensively managed and diverse grasslands were converted to seeded grasslands, often with only one type of grass, and in some cases also plowed into arable land [37]. Additionally, the landscape is characterized by a large number of small standing water bodies (kettle holes, consistently named "ponds" within the context of this paper), which were formed during the last glacial period. Ponds are defined as water bodies with a maximum extent of one hectare and cover about 5% of the arable land area over a total area of 38.000 km² in northeast Germany [38].

Samples were taken in three ponds (Fig 2) and four ditches (Fig 3) in October and November 2020. Pond 1 is entirely encompassed with trees, shrubs, and reed stands (*Phragmites* sp.). Ponds 2 and 3 are characterized by a mix of trees, shrubs, grasses (Poaceae), and reed (*Phragmites* sp.) in the riparian zone. Ditch 1 was dominated by reed (*Phragmites* sp.), while ditches 2–4 were dominated by grasses (Poaceae) in the riparian zone. See also Table 1 for descriptions of the water bodies and their surroundings.

## 2.2. Sampling and sample processing – invertebrates

A species protection exemption permit in accordance with Section 45 [7] No. 3 of the Federal Nature Conservation Act (BNatSchG) for the capture and killing of invertebrates with non-specific traps for agroecological studies in the Havelland district was issued by the Brandenburg State Office for the Environment (LfU).

Benthic invertebrates were sampled using hand netting (mesh size 500 μm) from the habitats (FPOM, CPOM, submerse macrophytes, reed) present in the selected water bodies. Individuals from all FFGs present were sampled and

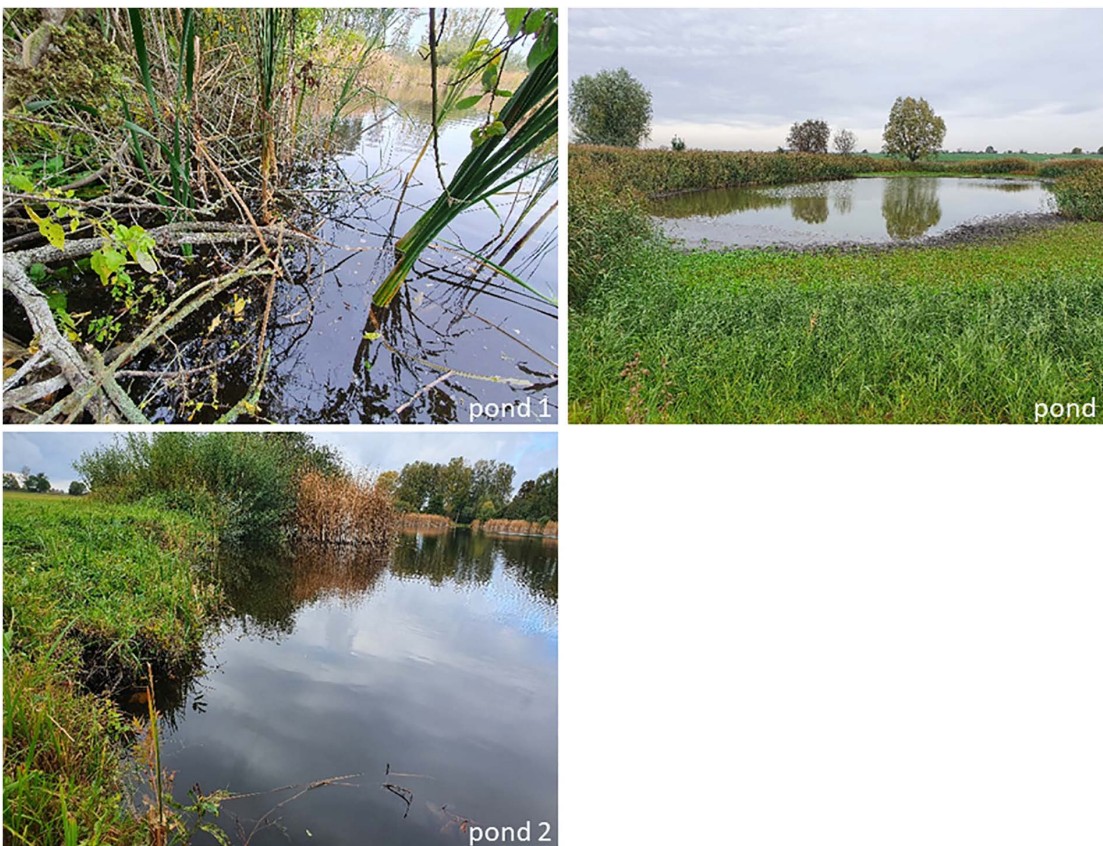

**Fig 2. Sampled ponds, pictures: Fee Nanett Trau, JKI.**

organisms were sorted on-site by morphology (to the level of order or family). Sampling was carried out until 10 specimens of the present orders/families were collected or as enough specimens were collected to create 10 pooled samples with the necessary sample weight as recommended by [30], whenever possible. Necessary sample weights were estimated by visual inspection in the field. The exact weighting in the tin capsules for analysis was carried out by the analysing lab, the Cornell Stable Isotope Facility. If 10 specimens were not found, a minimum of three samples was used for the analysis. All animals were kept alive for about 24 hours so that they can empty their gastrointestinal tract before analysis and then killed as quickly and humanely as possible with boiling water to avoid altering the isotopic signals through the use of ethanol [39,40]. Afterwards the samples were frozen before further processing. In the laboratory, the benthic macroinvertebrates were further taxonomically determined to family (where the order was assigned on site) or genus level when possible and sorted into the FFGs based on the freshwater ecology database [41]. Gastropods, Bivalves, and Trichoptera were removed from their shell or case and parasites were removed from afflicted Heteropterans before further sample processing to only analyze the isotopic signal of the body tissue of selected animals. In the next step, the samples were dried (48 hours at 60 °C, Memmert UFE 600). If the animal weight was > 2 mg (Sartorius LA 230 P), one individual organism was grounded into a powder. For lower sample weights, several individuals from one family were pooled into one sample to reach the optimum sample weight of 2 mg and then grounded. Material was homogenized using a mixer mill (Retsch MM400) and stored in Eppendorf tubes (1.5 ml). For each taxon identified, a maximum of 10–15 samples was analyzed.

The encapsulation as well as the isotopic analysis were carried out by the stable isotope laboratory at Cornell University (COIL), Department of Ecology and Evolutionary Biology, USA. Stable isotope analysis of $^{15}$N and $^{13}$C was performed

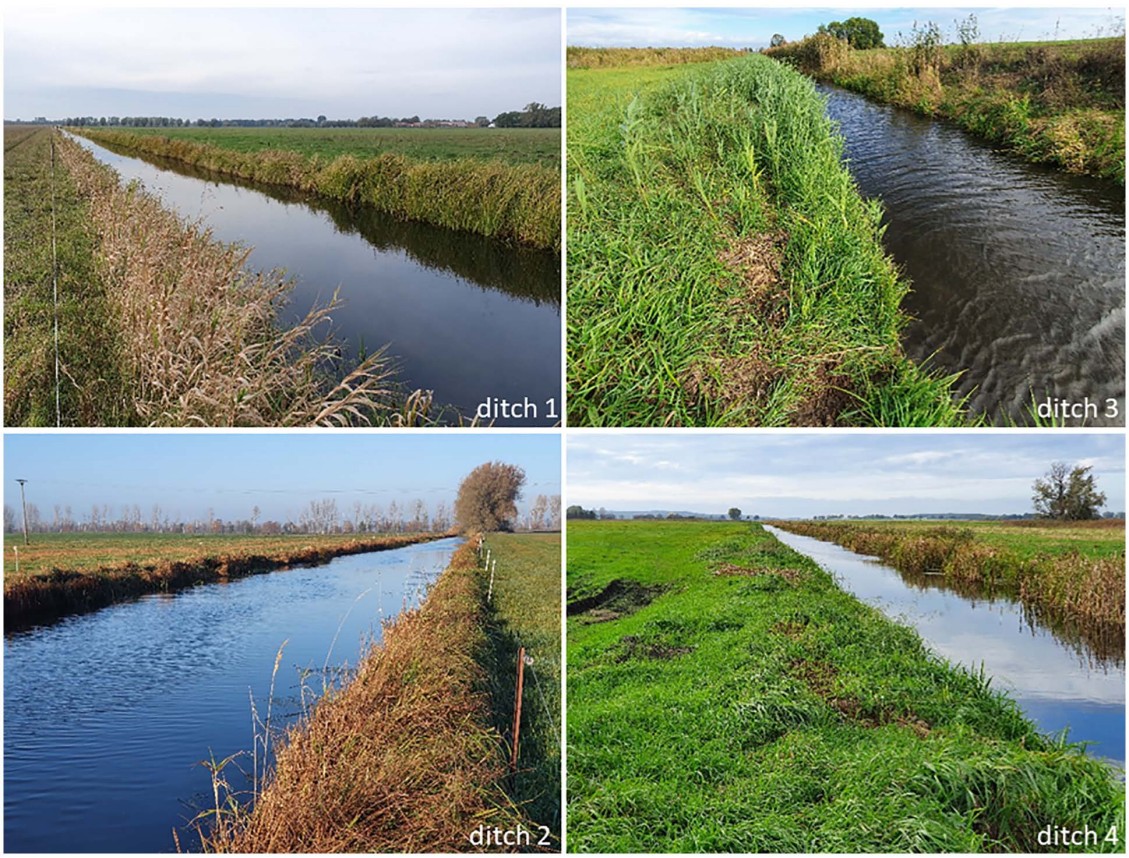

**Fig 3. Sampled ditches, pictures: Fee Nanett Trau, JKI.**

**Table 1. Description of the sampling sites. Habitats were described for the water bodies when present in the wadeable area of the shore length with a share < 5%.** The riparian zone is defined as the area from the edge of the field to the waterline where vegetation was estimated as a percentage. CPOM = coarse particulate organic matter; FPOM = fine particulate organic matter. Dominant types of riparian vegetation are written in bold.

| Sampling site | Habitats | Agricultural cultivation | Riparian zone [in %] | | | | |
|---|---|---|---|---|---|---|---|
| | | | Grasses | Herbaceous perennials | Trees | Shrubs | Reeds |
| Pond 1 | Reeds, CPOM, macrophytes | Grassland | 0 | 0 | 25 | 25 | **50** |
| Pond 2 | Reeds, CPOM, macrophytes | Grassland | 30 | 0 | 5 | 5 | **60** |
| Pond 3 | Reeds, CPOM, FPOM | Maize, cereal | 0 | 20 | 10 | 10 | **60** |
| Ditch 1 | Reeds, FPOM | Grassland | 20 | 0 | 0 | 0 | **80** |
| Ditch 2 | CPOM, macrophytes | Grassland | **100** | 0 | 0 | 0 | 0 |
| Ditch 3 | Reeds, CPOM, FPOM | Grassland | **80** | 0 | 0 | 0 | 20 |
| Ditch 4 | Reeds, FPOM, macrophytes | Grassland | **100** | 0 | 0 | 0 | 0 |

using a Thermo Delta V isotope mass spectrometer (IRMS) interfaced to a NC2500 elemental analyzer. Raw data, sample weights and quality control values for presented data can be obtained from the PANGEA repository [42,43]. Isotope corrections were performed using linear regression of all δ values using two additional in-house standards [44]. Results were reported as δ values as ratios of $^{13}C:^{12}C$ relative to Vienna PeeDee Belemnite and $^{15}N:^{14}N$ relative to atmospheric air (Eq. 1). Measurements were carried out from April 2021 to July 2022 in four batches.

$$\delta X = [(R_{sample}/R_{standard}) - 1] * 10^3 \tag{1}$$

Where X is $^{13}C$ or $^{15}N$, R is the corresponding ratio $^{13}C/^{12}C$ or $^{15}N/^{14}N$. Increased δ values show an increase of the heavy isotope in the sample [45].

## 2.3. Sampling and sample processing – water

Environmental chemical parameters were measured to identify potential variables that could explain the differences between the water body types. Water samples were taken via grab sampling. A mixed water sample of approximately five liters was taken from four to six locations around the shoreline of the ponds or along a stretch of 50 m from the ditches.

Environmental water parameters [oxygen content, electrical conductivity (EC), pH, and temperature] were measured *in-situ* once during sampling in the mixed water sample using a portable meter (WTW Multi 3430 IDS with OxiCal-SL FDO 925, TetraCon 925 and SenTix 940 probes, Germany). Samples for pesticide analysis were partitioned into two 500 mL glass bottles, with one bottle holding 50 mL of dichloromethane for pyrethroid analysis. The water samples were analyzed using multi-methods validated for the analysis of 99 active substances (S1 File).

Except for the pyrethroids, the selected active substances in the water samples were quantified by liquid chromatography mass spectrometry (LC-ESI-MS/MS) using the internal standard method after enrichment and clean-up via solid phase extraction (Chromabond HR-P, Macherey-Nagel, Germany, 50-100 µm, 3 mL, 200 mg). For LC-MS/MS analysis a 1290 Infinity II LC system (Agilent, USA) coupled to a QTRAP 6500 + mass spectrometer (SCIEX, USA) was used.

After concentration by liquid-liquid extraction of the sample, pyrethroids were analyzed by gas chromatography-mass spectrometry (GC-MS) using a Trace GC Ultra coupled to a TSQ Quantum GC XLS- mass spectrometer (Thermo Electron Corporation, USA). All pesticide analysis were done accordingly to [11].

Samples for nutrient analysis were filtered through a 0.45 µm cellulose acetate filter before analysis. The concentrations of $NH_4^+$ + and $PO_4^{3-}$ were measured photometrically using a spectrophotometer (DR/2000 Hach) according to DIN 38 406–5 and EN ISO 6878:2004, respectively. $NO_3^-$ and $NO_2^-$ concentrations were analyzed by high-performance liquid chromatography (HPLC, Agilent 1100 Series HPLC System DAD, USA).

### 2.3.1. Toxicity calculations.
To calculate the toxicity of the analyzed water samples for invertebrates and algae based on measured pesticide residue concentrations, the toxic pressure for each sample is calculated as the sum of toxic units; $TU_{sum}$ (Eq. 2). Toxic units were calculated for the pesticide concentrations based on half-maximal effective concentration (EC50) or 50% lethal concentration (LC50) values for invertebrates and algae. Analyzed pesticide concentrations were divided by the concentration for the most sensitive test organism and based on 48 h EC50 values for *Daphnia magna* or 96 h LC50 values for *Chironomus* sp. for the invertebrate toxicity. Toxicity calculations for algae are based on 72 h EC50 values for mostly, *Raphidocelis subcapitata (*syn: *Pseudokirchneriella subcapitata*) and *Scenedesmus subspicatus*. Data on the EC50 and LC50 values was obtained from the Pesticides Properties Database [46] or Ecotox database [47]. See also S1 File for details on substances measured and the test organisms.

$$TU_{sum} = \log \sum_{i=1}^{n} \left( \frac{C_i}{EC_{50_i}/LC_{50_i}} \right) \tag{2}$$

$C_i$ is the concentration of pesticide *i* in µg/L. Concentrations below the LOQ but above 0 were replaced with half of the LOQ.

## 2.4. Description of the dataset

The dataset contained $\delta^{13}C$ and $\delta^{15}N$ data covering six FFGs (Table 2) from 13 orders (see S5 File for details). FFGs were assigned to the individuals according to the family determination, if available, otherwise by order [33,41,48,49]. Classified

**Table 2. Number of samples available and used for analysis for each water body and the FFGs.**

| FFG | Pond 1 | Pond 2 | Pond 3 | Ditch 1 | Ditch 2 | Ditch 3 | Ditch 4 | Sum FFG pond/ditch |
|---|---|---|---|---|---|---|---|---|
| **Collector/filterer** | 0 | 0 | 0 | 3 | 0 | 0 | 9 | **0/12** |
| **Collector/gatherer** | 13 | 5 | 3 | 11 | 10 | 10 | 10 | **21/41** |
| **Grazer/scraper** | 37 | 8 | 11 | 16 | 14 | 18 | 16 | **56/64** |
| **Omnivore** | 15 | 31 | 22 | 28 | 23 | 30 | 25 | **68/106** |
| **Predator** | 19 | 19 | 14 | 38 | 45 | 26 | 60 | **52/169** |
| **Shredder** | 24 | 7 | 0 | 18 | 21 | 10 | 12 | **31/61** |
| **Sum type** | Ponds = 228 | | | Ditches = 453 | | | | |

*as collector/filterer* are Bivalvia and the family Culicidae, which filter suspended FPOM from the water column [41]. The families of Ephemeroptera and some dipterans belong to the group *collector/gatherers*, which feed on sedimented FPOM. *Grazers/scrapers* feed on algae tissue or biofilm growing on surfaces, also they feed on algae or plant tissue [41]. In the present dataset *grazers/scrapers* are mostly gastropods, e.g., of the families Lymnaeidae or Planorbidae. Within the group *omnivore*, families are gathered that could not be assigned to one distinct group for a number of reasons, such as species with different feeding groups in the family or due to the assignment of the family to the group *omnivore* by the database. In the present dataset Chironomidae, Corixidae, Haliplidae and Stratiomyidae were assigned as *omnivore*. *Predators* hunt on prey and examples are Dytiscidae, Ceratopogonidae or Coenagrionidae. Typical *shredders* are crustaceans like Gammaridae or Asselidae, which feed on plant tissue, fallen leaves or CPOM [41]. A detailed list of the families belonging to the FFGs can be found in the table S5 File.

## 2.5. Statistical methods

All data processing and statistical analyzes were performed with R Studio version 4.3.1 and 4.3.3 [50,51] and JAGS version 4.3.1.

**2.5.1. δ15N correction.** The influence of organic fertilizer was observed in ditches by examining the $\delta^{15}$N values of benthic macroinvertebrates (consumers). The $\delta^{15}$N values for ditches are higher than in ponds due to higher $\delta^{15}$N values in resources present in these water bodies (S2 File). To compare the FFGs in the two water body types this was corrected as follows: For both water body types the mean $\delta^{15}$N value for the consumers was calculated. The means were subtracted from each other, and the difference was subtracted from the $\delta^{15}$N values of the consumers in ditches only.

**2.5.2. δ13C correction.** The $\delta^{13}$C values for one pond are higher due to the influence of maize cultivation (C4-plant) on the consumers (Fig 4). For ponds 1 and 2 (no maize influence) and pond 3 (maize influence) the mean values for $\delta^{13}$C consumers were calculated. The means were subtracted from each other, and the difference was subtracted from $\delta^{13}$C values from consumers in pond 3 only.

**2.5.3. SIBER (Stable Isotope Bayesian Ellipses in R).** The R package SIBER [52] was used for the analysis of isotopic niche width of the FFGs and the comparison of food web structures between different water body types. The analysis follows a Bayesian approach to overcome uncertainties when calculating and comparing metrics [52]. The estimation by Bayesian inference incorporates error from the sampling process and allows robust comparisons between groups with different and small sample sizes [30].

First the raw data was plotted for the six FFGs (*collector/filterer, collector/gatherer, grazer/scraper, omnivore, shredder,* and *predator*) and the respective water body types (S4 File). Then the Layman metrics and the Bayesian inference-derived Layman.B metrics were calculated for the combination of water body and FFGs as group (37 groups) and the water body types pond and ditch as community as well as for the FFG as group (six groups) and the water body types pond and ditch as community. The Layman metrics are based on [24] and summarize information on the distribution of data points in the

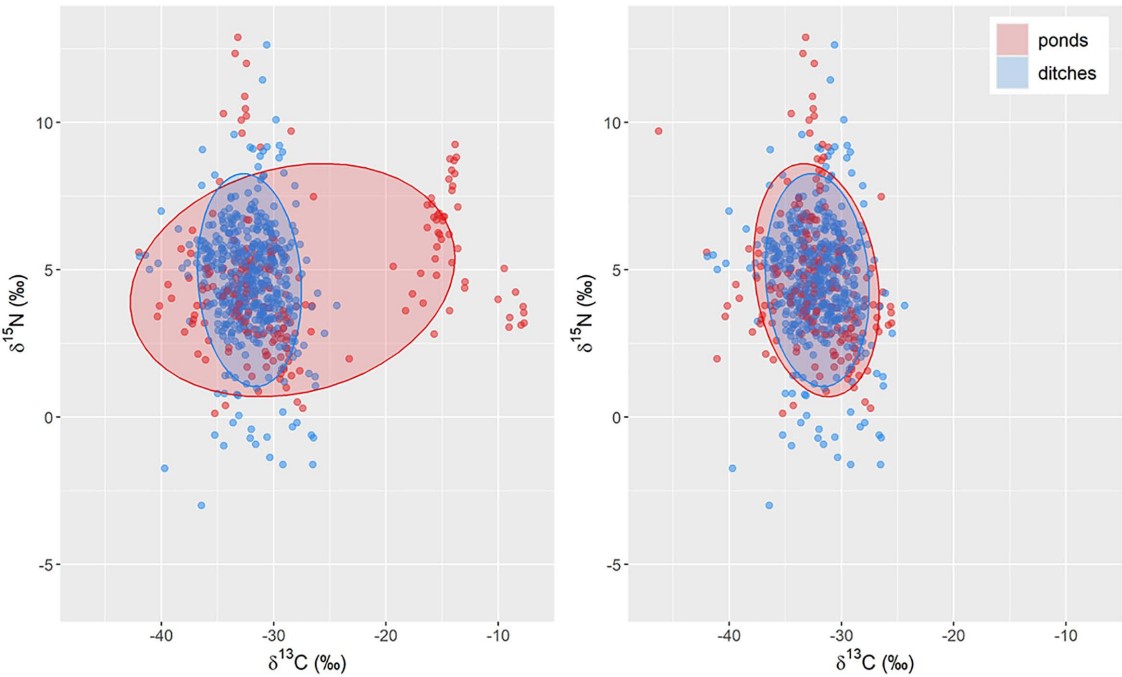

**Fig 4. Biplots for the macroinvertebrate data.** The left plot is corrected for δ15N influence of organic fertilizers on ditches. The right plot is corrected for δ15N as well as δ13C values for the influence of C4-plant maize on pond 3. Ellipses contain 40% of the data.

isotope space. The six metrics are the δ13C range, the δ15N range, the total area of the convex hull (TA), the mean distance to centroid (CD), the mean nearest neighbor distance (MNND) and the standard deviation of the nearest neighbor distance (SDNND) [30]. The TA describes the niche width, CD provides information on niche width and species spacing, the MNND is a description of density and clustering of species, and the SDNND gives information on the evenness of the species density [30]. To compare the groups the Welsch's t-test was used (confidence interval 95%). For the dbRDA (see below), the Layman metrics for the seven water bodies as community and FFG as group (six) were calculated.

Additionally, the Bayesian standard ellipse areas (SEA.B), corrected standard ellipse areas (SEAc), and convex hull area (TA) based on Bayesian interference were calculated for the FFGs in ponds and ditches. The Bayesian estimates provide robust measures of probability and can reflect model uncertainty by averaging many possible models into one single model [53]. TA is the total convex hull area enclosing all data points and thus showing the full data range. SEA.B is the Bayesian version of the standard ellipse area, usually compromising 40% of data points, providing a robust estimate with credible intervals that account for uncertainty [30]. The SEA.c based on SEA.B but is adjusted for small sample sizes, a frequentist measure of the core niche that gives a single point estimate [30]. Larger SEA values indicate more diverse isotopic signals and broader niches. Corrected δ15N and δ13C values were used to compare FFGs via SEA.c. The data was plotted using the function *siberDensityPlot* [54]. Both corrections were applied to eradicate the overlying influence of fertilization in ditches and the maize effect in one pond, allowing us to set the focus on the FFGs and to investigate whether differences among FFGs beyond these obvious factors.

The probability that the posterior distribution is smaller or larger for a group can be estimated by calculation of the SEA.B niche overlap. SEA.B overlaps between ponds and ditches were calculated for the FFGs based on the maximum likelihood estimates of the means and the covariance matrices of each group (function: *maxLikOverlap* calculated for the proportion of both overlaps, p-interval of 0.95). The calculation of the Bayesian Layman metrics (Layman.B) was

performed for the data set with corrected $\delta^{15}N$ values, as this was a general phenomenon observed for all ditches and changes trophic position in comparison to ponds and thus could override effects of for example pesticides. Maize cultivation was not a general phenomenon like fertilization in the study area, thus the original $\delta^{13}C$ values were kept for the following calculations. In other words, the real C-source (resource use) should not be artificially changed because of the niche width, but the N-source had to be scaled to the same level for the two water body types for comparability of trophic position without possible overriding effects of fertilization.

Bayesian models were run with JAGS and model parameters were defined according [30] to fit the ellipses.

**2.5.4. Multivariate statistics.** Environmental variables were standardized using the *scale* function of the *base* package [51], whereby the mean value was set to zero and scaled to unit variance. Variable selection for the dbRDA was carried out by the *bioenv* function from the *vegan* package [55]. This function detects the best environmental variable subset in a way such that the environmental variables show the maximum rank correlation (method: Spearman) with the Layman metrics matrix. The dissimilarity index for the Layman metrics matrix used was "mahalanobis", a measure that accounts for correlations across multiple variables. Selected environmental variables by the *bioenv* function, excluding the variables EC and $O_2$, due to small sample size of seven water bodies and the natural expected variability of the variables during the course of the day, were used for the dbRDA, which can show if an explanatory variable has significant impact on the response variables and how much of the variation is explained by each axis. (function *dbrda*, package *vegan*, 55). Selected variables were: nutrient concentration sum, number of pesticide detections, $TU_{sum}$ for algae, and $TU_{sum}$ for invertebrates. To test for the significance of the model, a permutation test on terms and axes was carried out ($p \leq 0.05$; function *anova.cca*, package *vegan*, 55). The calculation of the distance-based Redundancy Analysis (dbRDA) was performed for the data set with corrected $\delta^{15}N$ values (see explanation above).

# 3. Results

The raw data on the stable isotope composition ($\delta^{13}C$ and $\delta^{15}N$) of the consumers and sampled resources are available via the PANGEA repository [42,43]. An average of 76 isotope samples was analysed in ponds and 113.3 isotope samples in ditches (Table 2). In S5 File, an overview of the sampled FFGs and determination individuals analysed in pond and ditches is given. During the sampling campaign, 22 and 29 families were found in ponds and ditches, respectively.

## 3.1. Isotopic niches of communities in ponds and ditches

After correction for the influence of organic fertilizers and of maize, the $\delta^{13}C$ and $\delta^{15}N$ values of consumers show a similar range in ditches and in ponds (Fig 4 and S2 File). The biplot shows that ponds have a slightly larger ellipse area but that the water body types are comparable in general (Fig 4).

The posterior mode of the $\delta^{15}N$ range is comparable between the two water body types (after correction for organic fertilization in ditches) (Fig 5a). The typical range of $\delta^{13}C$ values is significantly distinct in the two water body types and larger in ponds (Fig 5d). The posterior distribution of TA and thus total niche width and the spacing of individuals (mean distance to centroid, CD) are significantly larger in pond communities compared to communities in ditches (Fig 5b, 5e). The posterior values for the MNND (Fig 5c) and thus clustering of individuals within the community is larger in ponds, whereas the evenness of density is similar in ponds and ditches (Fig 5f).

For an illustration of the Layman.B metrics calculated for both stable isotopes corrected, see also figure S3 File.

## 3.2. Comparison of functional feeding groups

In ditches, a mean value of 44.5 isotope samples was recorded for primary consumers (*collector/filterer, collector/gatherer, grazer/scraper, and shredder*) and a mean value of 26.5 isotope samples of *omnivores* and a mean value of 42.25 isotope samples of *predators* was recorded (Table 2). In comparison, a mean value of 36 isotope samples for primary consumers

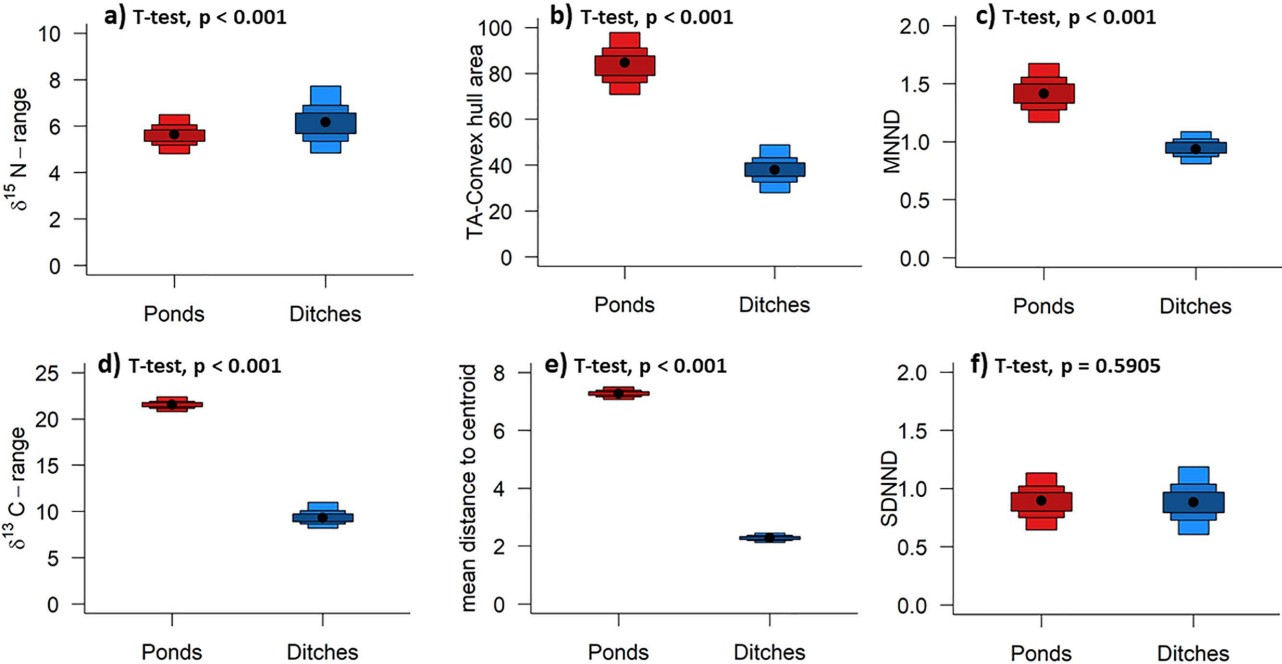

**Fig 5. SIBER density plots for the Layman metrics computed via Bayesian statistics, based on the dataset corrected for δ¹⁵N. Black dots show the modes.** Shaded boxes represent the 50%, 75%, and 95% confidence intervals, from dark to light red/blue. Grouping is based on a combination of the individual small water body (7 SWB) and the six FFGs (in total 37 groups), the communities are assigned to the SWB types (pond, ditch). δ¹⁵N values of the ditches are corrected for the influence of organic fertilizers.

(*collector/filterer, collector/gatherer, grazer/scraper, and shredder*), 22.6 isotope samples for *omnivores* and 17.3 isotope samples for *predators* was recorded in ponds (Table 2).

There are observable differences for the SEA.B modes and confidence intervals between FFGs when comparing ponds and ditches. SEA.B and SEAc modes for *shredders* are higher in ditches than in ponds (Fig 6). The groups *omnivore* and *grazer/scraper* have larger SEA.B/SEAc modes in ponds (Fig 6). Values for *predators* and *collector/gatherers* are similar in the two water body types (Fig 6). The FFG *collector/gatherer* was dominated by ephemeropterans (*Cloeon* sp.) in both water body types (Fig 6) which explains the similarity of SEA.c modes and shows that this FFG is similar in its resource use in both water body types (Fig 6). The credible intervals for *predators* in ponds and ditches are rather small, which shows that the uncertainty is low and that the dataset provides a precise representation of the sample mean for this group. *Predators* were in general observed with more taxa groups in ditches. Six orders (Coleoptera, Diptera, Heteroptera, Hirudinea, Megaloptera, Odonata) and 13 families and were recorded there, while five orders (Coeloptera, Diptera, Heteroptera, Odonata, Trichoptera) and eight families were recorded in ponds (S5 File).

SEA.B niche overlaps for the FFGs between ponds and ditches vary between 15 and 22.9% (corrected only for δ¹⁵N) and between 22.9 to 44.7% when the data is corrected for δ¹⁵N and δ¹³C (Table 3). For *shredders* the overlap is the same for both correction approaches, as there were no *shredders* found in the maize pond and thus no δ¹³C correction was needed. For the corrected for δ¹⁵N and δ¹³C values, niche overlaps for *predators* (44.7%) are largest in ponds and ditches. *Shredders* show least similarity in niches between ponds and ditches (22.9%).

### 3.3. Effects of agricultural stressors on the observed results

Results for the environmental parameters, nutrient concentration and toxicities of the water samples measured to describe the quality of the water bodies (Table 4).

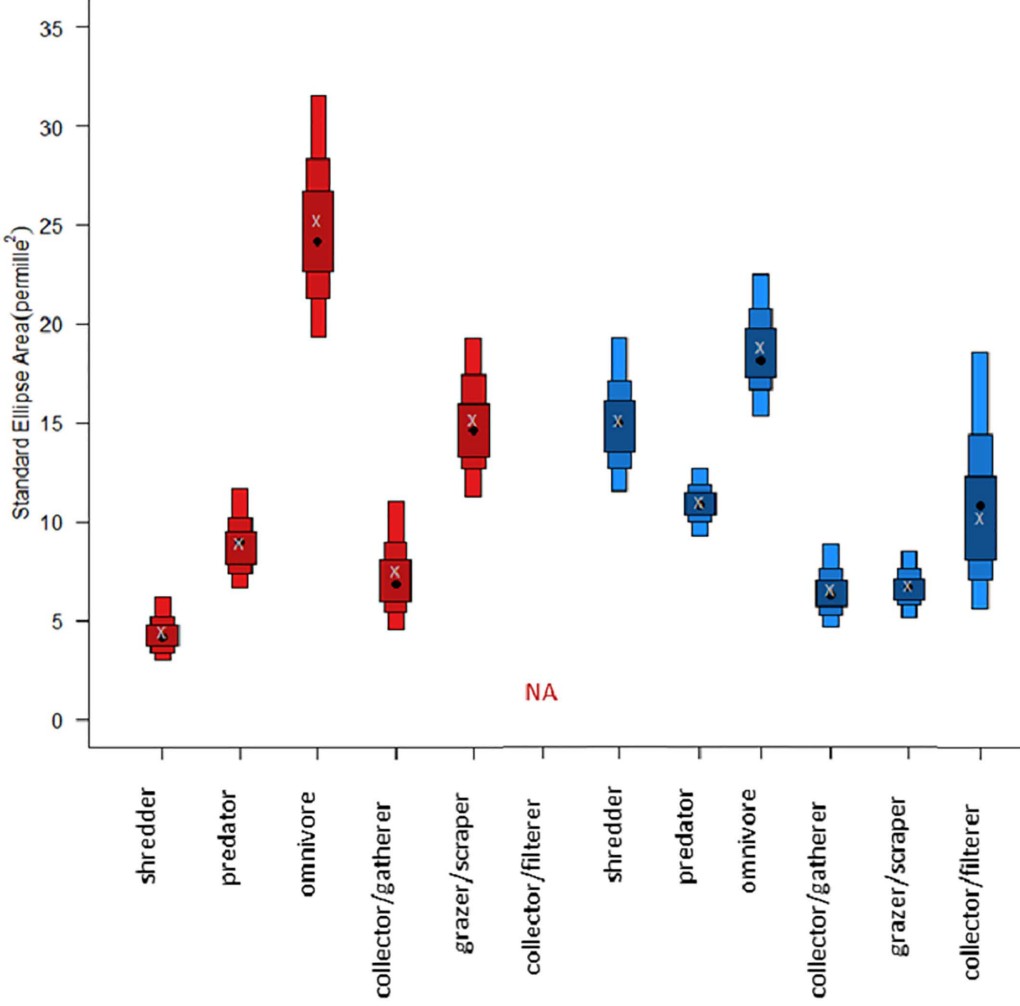

**Fig 6. SIBER density plots computed via Bayesian analysis using corrected δ$^{15}$N and δ$^{13}$C values.** Plots are shown for the FFGs in ponds (red) and ditches (blue). Black dots are the modes of SEA.B. The grey X indicates the size-corrected standard ellipse area (SEAc) mode. Shaded boxes represent the 50%, 75%, and 95% confidence intervals.

**Table 3. Niche overlaps of ponds and ditches for the functional feeding groups (FFGs) calculated for two datasets.** The overlap for collector/filterer cannot be calculated due to their absence in pond samples.

| Functional Feeding Group | Niche overlaps of ponds and ditches in %, corrected for N values | Niche overlaps of ponds and ditches in %, corrected for C and N values |
|---|---|---|
| Collector/gatherer | 15.0 | 28.4 |
| Grazer/scraper | 16.9 | 29.8 |
| Omnivore | 18.6 | 38.6 |
| Predator | 20.0 | 44.7 |
| Shredder | 22.9 | 22.9 |

**Table 4. Overview of the measured water quality and environmental parameters for the seven water bodies. Values for NO$_2^-$ [mg/L] not shown as they are all 0. EC = electric conductivity, Temp. = water temperature.**

| Sampling site | Temp [C °] | EC [μS/cm] | pH | O$_2$ [mg/L] | NH$_4^+$ [mg/L] | NO$_3^-$ [mg/L] | PO$_4^{3-}$ [mg/L] | NH$_4^+$ +NO$_3^-$ [mg/L] | TUsum algae | TUsum invertebrates | No. of habitats | No. Pesticide detections |
|---|---|---|---|---|---|---|---|---|---|---|---|---|
| Pond 1 | 7.4 | 1570 | 6.2 | 5.2 | 0.01 | 0.20 | 0.25 | 0.21 | −2.686 | −5.221 | 3 | 3 |
| Pond 2 | 10.5 | 1112 | 6.5 | 10.6 | 0.09 | 0.14 | 0.05 | 0.23 | −4.057 | −5.074 | 2 | 1 |
| Pond 3 | 11.2 | 881 | 7.0 | 13.8 | 0.08 | 0.23 | 0.11 | 0.31 | −2.650 | −4.632 | 3 | 4 |
| Ditch 1 | 9.7 | 588 | 7.0 | 11.5 | 0.86 | 1.10 | 0.44 | 1.96 | −2.685 | −5.428 | 2 | 4 |
| Ditch 2 | 8.3 | 815 | 7.6 | 6.1 | 0.08 | 0.72 | 0.06 | 0.80 | −3.253 | −5.116 | 1 | 6 |
| Ditch 3 | 11.0 | 928 | 7.0 | 8.8 | 0.08 | 0.36 | 0.06 | 0.44 | −2.614 | −4.817 | 3 | 7 |
| Ditch 4 | 12.7 | 900 | 8.0 | 8.8 | 0.08 | 1.32 | 0.03 | 1.40 | −3.456 | −4.925 | 2 | 7 |

Water quality parameters measured in the water samples indicate a lower number of pesticides in ponds (mean = 2.7, SD = 1.5) and higher numbers of pesticides in ditches (mean = 6.0, SD = 1.4). The majority of the total pesticides detected (ponds and ditches) are herbicides, accounting for 81%, while 19% are fungicides. Furthermore, nutrient concentration sum is higher in ditches (mean = 1.15, SD = 0.70 mg/L) compared to ponds (mean = 0.25, SD = 0.05 mg/L). However, the toxicity of the samples for algae and invertebrates is similar in ponds and ditches. Dissolved oxygen concentrations vary from 5.2 mg/L to 13.8 mg/L in ponds and from 6.1 mg/L to 11.5 mg/L in ditches. EC varies from 881 to 1570 μS/cm in ponds and from 588 to 928 μS/cm in ditches.

The dbRDA plot does not show a clear pattern for ponds and ditches (Fig 7). The isotopic composition of benthic invertebrates and the resulting Layman metrics of ponds 3 and 2 are influenced by the toxicity of water samples for invertebrates (TUsum_invert). Pond 1 and the ditches, except ditch 2, are influenced by the toxicity of water samples for algae (TUsum_algae), the concentration of nutrients (nutrients_sum), and the number of pesticides found (N_pesticides). Constrained variables at the first axis explain 75% of the variance. However, the permutation test resulted in no statistical difference (p ≤ 0.05) for the tested model, model parameters, or axes.

## 4. Discussion

The corrections for the influences of fertilizers (δ$^{15}$N) and maize (δ$^{13}$C) were applied to eradicate the overlaying effects, allowing us to set the focus on the differences between either the two water body types or the FFGs in the two water body types and their niche overlaps. In order to investigate whether differences among FFGs niches beyond the obvious factors (fertilization, maize) can be detected we corrected for both isotopes. For comparison of the two water body types only corrections of δ$^{15}$N (fertilizer) were applied. Thus, after the correction of δ$^{15}$N for the influence of fertilizer, which was clearly visible in consumers in ditches, we were able to comparatively assess differences in the two water body types. Such relative enrichment was also found by [56] in urban and agricultural streams compared to woodland streams. However, within the agricultural affected sites studied here, isotopic niches could not depict benthic invertebrate community changes due to the agricultural stressors of nutrient input and pesticides. Exceptions are the general impacts on basal C and N resources (e.g., by fertilization or maize/C4 cropping), which were observed in the isotope niches. Also, the comparison of the isotope niches of the two water body types, pond and ditch, did not reveal any clear patterns, relating to the agricultural stressors, when analysing the FFGs.

The hypothesis that we expected simplified trophic structures in ditches due to agricultural activity and low habitat variability can only be partially confirmed. The finding that the likely distribution of total niche width and distances between the individuals is greater in pond communities than in ditch communities indicates that resource diversity (that may be driven by the amount or presence of different habitats) is higher in ponds, which led to more extreme isotope ratios [32]. Furthermore, consumers are more divergent in ponds in terms of the isotopic niche MNND, while the evenness of density

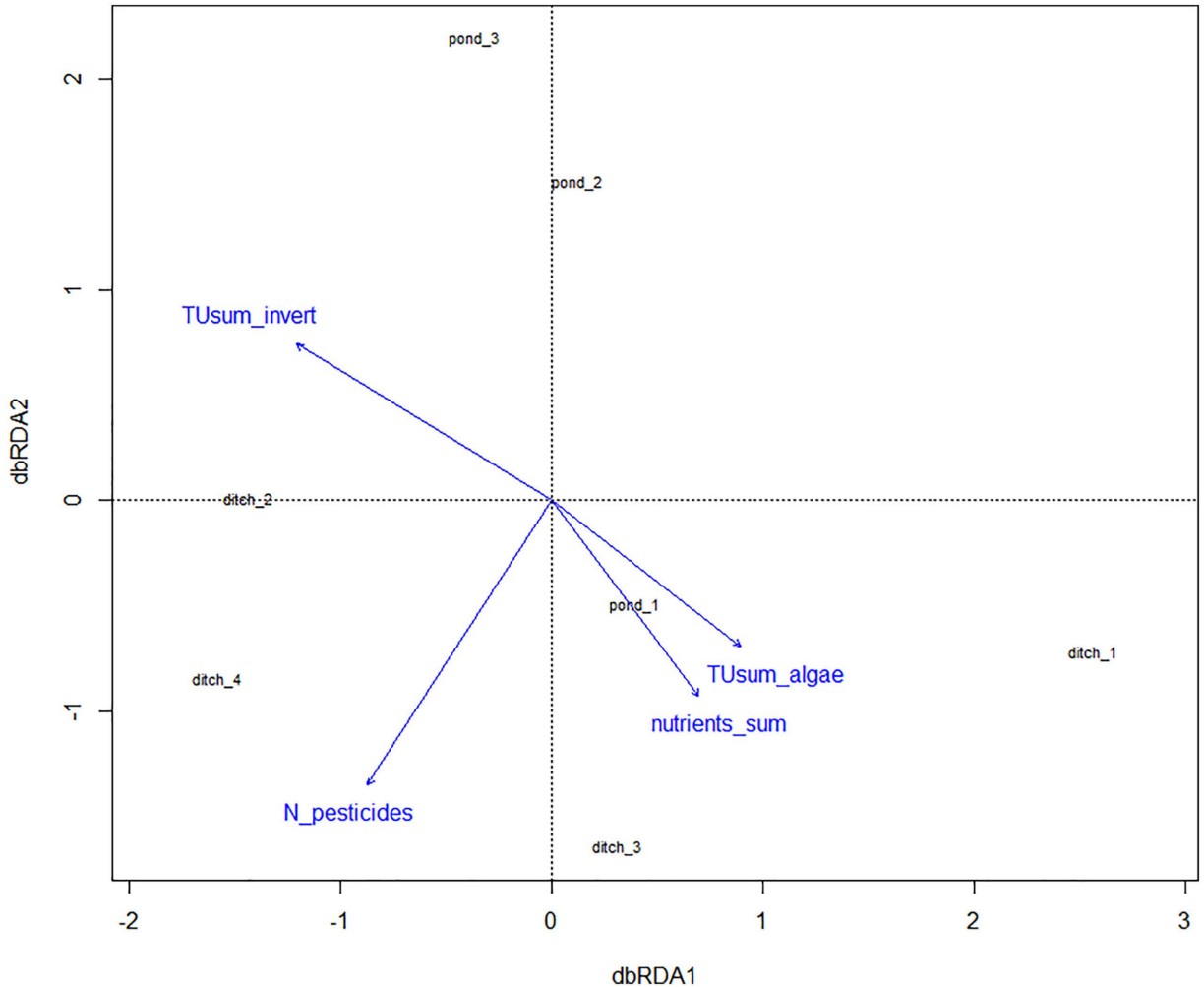

**Fig 7. dbRDA plot, distance measure = Euclidean.** Water bodies are plotted according to the Layman metrics calculated for the stable isotope results of the communities. Environmental parameters with significant correlations are plotted in blue. The length of the arrows shows how much the variable correlates with the water bodies.

SDNND is similar in ponds and ditches. This indicates less clustering in pond communities at the same evenness level, which in turn indicates more available resources and/or habitats in ponds that were evenly utilized and colonised [57].

Contrary to our expectations, the FFG *predator* was more diverse in ditches, we identified more families in ditches. The high number of *predators* in samples in ditches (mean value of 42.3 in ditches and mean value of 17.3 in ponds) could be explained by the availability of a high number of primary producers facilitated by high nutrients inputs and thus high primary production, which leads to more available prey [58,59]. This was underlined by the results, which showed that the potential number of prey was higher in ditches (mean primary consumers = 44.5N and mean *omnivores* = 26.5N), compared to ponds (mean primary consumers = 36.0N and mean *omnivores* = 22.6N). However, we have to be careful with this conclusion as we did not measure primary production. Alternatively, *predators* could be sustained by the brown food web to a higher extent in ditches (mean *shredder* = 61.0N) than in ponds (mean *shredder* = 31.0N). This would also be in line with an earlier modelling study indicating that *predators* may rely on energy derived from the brown food web but also underlines that brown and green food webs are interacting and cascading effects are theoretically in place [60]. This could

be especially true for non-natural sites, as a study by [61] found that *predators* rely on detritivores more in restored sites than in reference sites. The influence of changes on the base of the food web propagated to higher trophic levels was also observed by [58], where nutrient pollution promoted biofilm production and led to an increase in herbivory and carnivory by *omnivores*. This positive influence of fertilization might be explained by intermediate but not high nutrient enrichment, as in the case of [62], which found highest densities of invertebrates at intermediate levels of nutrient enrichment (primarily of phosphate). The high taxa number and number of mean isotope samples is probably stimulated by bottom-up effects of nutrient enrichment [56,63]. In contrast to that, according to [58], the effects of long-term nutrient enrichment should lead to a reduction in species numbers and shift community structures. Although we did not directly measure species numbers, we did not observe this phenomenon in the ditches compared to the ponds, as the number of families and individuals found is greater in ditches even though the nutrient concentrations were higher in ditches.

However in contrast to results from [58], the nutrient pollution in ditches did not lead to less diverse communities for the FFG *predator* in ditches compared to ponds, instead the opposite was the case. In ditches, the number of recorded families belonging to the group of *predators* was nearly twice as high compared to ponds. Therefore, functional redundancy for this feeding type can be assumed [64] which is necessary to ensure resilient systems [65].

The *omnivore* FFG expressed a variety of feeding strategies. In pond 2 they were clearly mostly predatory with two clusters formed (S4 File). One cluster formed by individuals with the highest $\delta^{15}N$ values in that water body and a second cluster with the $\delta^{15}N$ values on the same level as the FFG *predators* in that pond (S4 File). In the ditches 2, 3 and 4 they were predatory as well but $\delta^{15}N$ values were on the same level as the values of the *predators*. Some individuals classified as *omnivore* showed very low $\delta^{15}N$ values which indicates an herbivorous feeding preference. The *omnivores* FFG formed clusters on the upper and lower level of $\delta^{15}N$ values in four water bodies (two ditches and two ponds) which could show the shift of *omnivores* towards either herbivorous or predatory lifestyles as stated by [58]. However, according to [58], this shift occurs in polluted water bodies, thus also the pond communities might show signs of the influence of agricultural stressors. Nevertheless, responses of *omnivores* are dynamic, and shifts could also occur due to other perturbations, e.g., temporal shifts in resources across seasons.

The larger SEA.c mode for FFG *grazer/scraper* in ponds could be explained by a larger number of resources available in ponds, which had a more diverse habitat in and around the water body compared to the ditches in this study. A low diversity of littoral habitats was shown to be associated with low numbers of trophic links and less resource and consumer diversity [66], which might be the case in the ditches in our study [66]. This would mean that the nutrient-rich ditches are not diverse in resources. Additionally, the larger number of habitats (e.g., macrophytes, reeds, CPOM) and/or more resources used can also explain commonly observed patterns of higher taxa richness and the associated possible different preferences in ponds [7–9]. Two *grazer/scraper* taxa groups (Diptera, Gastropoda) and five families were found in ponds, but only one *grazer/scraper* taxa group (Gastropoda) and three families in ditches. In eutrophic waters, cyanobacteria blooms can occur which are toxic for other primary producers and can lower the number of *grazers*, but the concentration of cyanobacteria was high for one pond only (data not shown). This could be a measurement of a peak value that had already been exceeded in the other waters, as cyanobacterial blooms can occur very quickly.

The absence of a distinct difference, which is attributable to the stressors nutrient concentrations and pesticide residues, is also underlined by the dbRDA. The configuration of the water bodies shows that they are influenced individually by different parameters, and other factors which were probably not detected. In general, the composition of benthic invertebrates and the resulting Layman metrics appear to be influenced either at the production level (indicated by the variables nutrients sum, toxicity for algae, and number of pesticides detected (mainly herbicides)) or at the consumer level (indicated by the toxicity for invertebrates). Ponds 2 and 3 are similar to each other and are largely influenced on the consumer level. The four ditches and pond 1 are more strongly influenced at the primary producer level, so bottom-up effects are also probably stronger here, as the variables explaining those the best are all connected to primary production.

Nonetheless, the FFGs *omnivore* in ponds and *collector/filterer* in ditches show wide credible intervals that indicate a high degree of opportunism and variability in their feeding strategies which is consistent for the FFG *omnivore*. For the FFG *collector/filterer* further sampling would be needed to predict modes with less uncertainty.

The FFG *collector/filterer* was sampled with below three individuals in ponds thus the data was not used in the analysis. Also, in ditches their occurrence was low (10 Bivalvia, two Diptera, see S5 File). It is only possible to speculate about the reasons, but potential factors could be: i) time of sampling, low occurrence of *collector/filterer* in November, as their food base is low due to the reduced availability of suspended particles as a result of less algae growth and less agricultural activity and therefore less erosion, ii) hydrological conditions [67] and sedimentation [68] in ponds which are not favorable for, e.g., Bivalvia. As *collectors/filterers* were present in low numbers in both ponds and ditches (only in the ponds excluded from the analysis), we did not discuss this further.

Additionally, the time of sampling (autumn) has to be considered and has most probably influenced the detection of taxa and pesticides. Pesticide residues in autumn and winter are lower, as pesticides are mainly applied in spring and early summer due to the general agricultural practice to establish crops such as oilseed rape, maize or cereals during the main growing season [69,70]. Information on community composition may have been skewed toward a less diverse composition, with lower numbers of especially Ephemeroptera, Trichoptera, and Diptera as the larvae have already emerged from the water bodies to reproduce and lay eggs.

Although higher numbers of pesticides were found in ditches compared to ponds, this impact does not transport into the expected toxicity gradient for algae and invertebrates, presumably due to the application of low-toxicity pesticide products by the farmers. Additionally, [71] recently found patterns of synchronous toxicity levels towards algae and aquatic plants between water bodies with a comparable distance to each other like in our study. This should be accounted for in further studies, and future sampling between the months of April and June is generally recommended to sample larvae stages of merolimnic invertebrates and to match the period of pesticide applications [71,72]. Furthermore, the high explanatory power of 75% of the constrained variables on the first axis most likely resulted from running the model on a small dataset and with a high number of explanatory variables.

## 5. Conclusion

With the obtained dataset, we were unable to recognize distinguished differences of the effects of agriculture (nutrient and pesticide pollution) on isotopic niches of the benthic invertebrates. Agricultural activity and a homogenous environment did not lead to simplified structures in the ditches compared to ponds. Ecological systems respond in complex ways to the presence of abiotic and biotic stressors. Particularly small systems are unstable, and exposures are often unclear, leading to ambiguous responses. The individual water bodies are influenced by different parameters, and the impacts of individual structures within each water body are more pronounced than influences based on the type of water body, at least for the dataset present. Larger datasets are promising options to overcome this issue, which to our knowledge are not available for small water bodies in the agricultural landscapes in Europe. Additionally, selected water bodies should be monitored more closely and analyzed on a broader temporal scale to detect potential effects of agriculture on benthic invertebrate communities over time.

## Supporting information

**S1 File. Active substances (99), with their limits of quantification and detection, measured, as well as the endpoints and test organism used for the toxicity calculations.**
(DOCX)

**S2 File. Biplot for benthic macroinvertebrate before correction for $\delta^{15}N$ values from and the influence of organic fertilization.** Ellipses incorporate 40% of the data.
(DOCX)

**S3 File. SIBER density plots for the Layman metrics, computed via Bayesian analysis.** Black dots are the modes. Shaded boxes represent the 50%, 75% and 95% confidence intervals, from dark to light red/blue. Grouping is based on the FFG (six groups), the communities are assigned for the water body types (pond, ditch).
(DOCX)

**S4 File. Representation of the FFGs as coloured dots and ellipses, 40% of the data points are represented by the ellipses.**
(DOCX)

**S5 File. Overview of the sampled FFGs and determination for the ponds and ditches.**
(DOCX)

**S6 File. Layman metrics for ponds and ditches, calculated for the FFGs and the different water bodies as grouping.**
(DOCX)

**S7 File. Biplots for benthic macroinvertebrates in ponds (top) and ditches (bottom).** Raw data was corrected with $\delta^{15}N$ means of resources.
(DOCX)

**S8 File. Biplots for benthic macroinvertebrates in ponds (top) and ditches (bottom).** Raw data was corrected with $\delta^{15}N$ means of resources and $\delta^{13}C$ values were corrected for the influence of C4-plant maize.
(DOCX)

## Acknowledgments

The authors thank Marlen Heinz, Manuel König, Karin Meinikmann, Maren Stockmann, and Elke Zeidler for their assistance during the field sampling campaign. Furthermore, we thank Dominique Conrad, Michael Glitschka, Christine Reichmann, Ina Stachewicz, and Gabriele Smykalla for the sample preparation and pesticide and nutrient analysis. In addition, we thank Dominique Conrad and Jörg Stauch for their help in preparing the biological samples. Thanks also to Matthias Stähler for discussions on analytical methods and Mario Brauns, Kristin Scharnweber, René Sahm, and Matthias Pilecky for fruitful discussions on data analysis. Furthermore, we thank Holly Blevins for proofreading.

## Author contributions

**Conceptualization:** Fee Nanett Trau, Stefan Lorenz.

**Data curation:** Fee Nanett Trau, Kathrin Fisch, Stefan Lorenz.

**Formal analysis:** Fee Nanett Trau, Kathrin Fisch.

**Funding acquisition:** Stefan Lorenz.

**Investigation:** Fee Nanett Trau, Stefan Lorenz.

**Methodology:** Fee Nanett Trau, Kathrin Fisch.

**Project administration:** Stefan Lorenz.

**Resources:** Stefan Lorenz.

**Supervision:** Fee Nanett Trau, Stefan Lorenz.

**Validation:** Fee Nanett Trau, Kathrin Fisch.

**Visualization:** Fee Nanett Trau.

**Writing – original draft:** Fee Nanett Trau, Kathrin Fisch, Stefan Lorenz.

**Writing – review & editing:** Fee Nanett Trau, Kathrin Fisch, Stefan Lorenz.

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
