## [Decision Letter · Decision Letter 0]

25 Mar 2025

Dear Dr. Lorenz,

Thank you for submitting your manuscript to PLOS ONE. After careful consideration, we feel that it has merit but does not fully meet PLOS ONE’s publication criteria as it currently stands. Therefore, we invite you to submit a revised version of the manuscript that addresses the points raised during the review process.

**ACADEMIC EDITOR:**

We look forward to receiving your revised manuscript.

Kind regards,

Marina Vilenica

Academic Editor

PLOS ONE

3. Thank you for stating the following financial disclosure: [This work was funded by the German Federal Ministryof Food and Agriculture (BMEL) via the Agency for Renewable Resources (FNR; project number22012018) based on a decision of the Parliament of the Federal Republic of Germany.]. 

6. We note that[Figure 1 in your submission contain [map/satellite] images which may be copyrighted. All PLOS content is published under the Creative Commons Attribution License (CC BY 4.0), which means that the manuscript, images, and Supporting Information files will be freely available online, and any third party is permitted to access, download, copy, distribute, and use these materials in any way, even commercially, with proper attribution. For these reasons, we cannot publish previously copyrighted maps or satellite images created using proprietary data, such as Google software (Google Maps, Street View, and Earth). For more information, see our copyright guidelines: http://journals.plos.org/plosone/s/licenses-and-copyright.

Reviewers' comments:

Reviewer's Responses to Questions

**Comments to the Author**

1. Is the manuscript technically sound, and do the data support the conclusions?

Reviewer #1: Partly

Reviewer #2: Partly

2. Has the statistical analysis been performed appropriately and rigorously?

Reviewer #1: I Don't Know

Reviewer #2: Yes

3. Have the authors made all data underlying the findings in their manuscript fully available?

Reviewer #1: Yes

Reviewer #2: Yes

4. Is the manuscript presented in an intelligible fashion and written in standard English?

Reviewer #1: Yes

Reviewer #2: Yes

Reviewer #1: I finalized revising the manuscript “Assessing agricultural effects on benthic invertebrate communities in ponds and ditches using δ¹⁵N and δ¹³C isotope niches”. The topic is interesting and attractive to a broad audience. However, major improvements need to be done in the manuscript. The questions and hypotheses, and how the methods were applied to solve these questions, were rather cryptic for me. It was generally hard to understand from the introduction which is the knowledge gap the authors are trying to fulfill. Moreover, without clear questions, assessing the science behind the methods and results was difficult. My major misunderstanding is why the authors expect differences between the two types of ecosystems assessed, and therefore, why this is an interesting question in the framework of agricultural stressors. I think the authors have an interesting and valuable dataset with the potential of answering interesting ecological questions. However, significant work needs to be done. I left detailed comments for each of the sections.

Abstract

14: Why do ponds and ditches harbor particularly endangered or scarce taxa? Moreover, why do you mean by scarce, is it low abundant taxa or taxa with low occupancy? I’d recommend removing this last part of the sentence.

16 I disagree. Numerous studies have shown the adverse effects of agricultural land use in biodiversity. Perhaps what you mean is that these habitats are likely to face multiple stressors, all derived from agriculture, which results in a problematic complex to address. More importantly, the reader does not know so far which are these previous approaches and why they were not successful. For example, what do you mean by “classic structural biodiversity metrics”?

19: We analyzed the carbon and nitrogen stable isotope composition of the invertebrates and the pesticides and nutrient residues of water.

21 readers do not know what are Standard ellipse and Layman B community metrics. I recommend leaving technical terms out of the abstract.

22 FFG should be introduced earlier.

Which differences? You’re leaving the reader hanging on the unexpected differences but don’t tell them (us) which.

28-32 See my previous comment. These sentences are uninformative for readers. We don’t know why case-by-case patterns emerged, or where the differences lay. What do you mean by the production or consumption level? How was production measured or estimated? We also don’t know which strong fluctuations on parameters did you observed; therefore, we don’t know how did you reach this conclusion.

Remove In the end

Introduction

is it specialization, and use of fertilizer and pesticides homogenizing the landscape? I’d say that the expansion of agricultural land is what homogenizes the landscape. Ref 3
What do you mean by semi-natural habitats? Under less intensive agriculture?
Starting a paragraph with an example is quite counterintuitive. You haven’t introduced yet non-productive areas, so why are you giving us examples?
Same as in the abstract: would you define scarce more precisely.
a high richness compared to what? Also, these groups are highly abundant in many aquatic ecosystems; therefore, why are they particularly important in ditches?
I think this is your topic sentence for this paragraph; therefore, it should be the first one -or maybe the second one, but at the beginning!
but this contradicts your previous sentence in line 48
in which sense is challenging? Actually, invertebrates are one of the most used groups for biodiversity assessment because sampling is comparatively easy and cheap to other taxa, and they are very good indicators of ecosystem status and functions.
But there is extensive literature showing the effects of agriculture on biodiversity. I think you need to be more specific as to why functional approaches outperform, or complement, taxonomic approaches. For example, why would functional approaches be more sensible to persistent stressors or short gradients? Ref 8
why were these other traits not included in your analysis? One important effect of agricultural land use is the change in inputs of organic matter to the water bodies; this justifies the use of feeding groups. However, other effects are channelization or modification of the physical structure, which might produce a response in locomotion; changes in productivity (eutrophication), with effects on body size and respiration; changes in temperature, with effects on thermal tolerance and respiration. In other words, you need either to include more traits, or to justify why feeding groups are the best approach to address the effects of agriculture, including persistent stressors and short gradients.
this paragraph is disconnected with the rest of the introduction. While important concepts need to be defined, the reader wants to know how the niche concept(s) is related to your own work and very importantly, to your question and knowledge gap. It is your task, as writers, to tell the reader how this concept can help understanding effects of stressors in biodiversity -not give the reader homework . For example, I’d recommend combining this paragraph with the trait paragraph, since the niche is going to be defined based on those traits. This will provide more coherence between your paragraphs and shorten your introduction.
we used. You haven’t defined isotopic niche nor realized niche. Feeding resources? And which habitats? Do you mean between ponds and ditches? If so, why feeding resources and, therefore, isotopic niches would differ between them?
we compared
which groups?
are you assessing more than one stressor? Or is it pesticides the only metric you are using?
we asked. can
I don’t fully understand your questions. Are you assessing isotopic niches in a gradient of agricultural stress, as measured by pesticides? Why do you expect differences, or not, between ponds and ditches?
to answer these questions…
what is the difference between niche differentiation and niche specialization? And why reduced niche specialization would result in less overlap? I’d expect the opposite. How about the expectations for ponds? Do ponds have different effects of agriculture?

It is not clear to me which questions are being answered and how. Is it the question related to differences in niche width due to different levels of agricultural stress? Then, would these effects differ between ponds and ditches? Why? What do you mean by habitat variability? Is this variability inside each water body, or among types?

Methods

I think that saying During the 1980s, the wet lowlands… is enough. Moreover, I’d turn the sentence around: The wet lowlands in the Havelländisches region were drained … during the 1980s.

105: What do you mean by traditional grasslands? As in, they were already agricultural lands but transformed into monoculture crops? Or were they natural grasslands converted to agriculture?

What about ditches?
describe the site’s characteristics in order, from Pond 1 forward, and ditch 1 on.

Table 1: How were these habitats characterized? Do they correspond to the littoral zone, the whole water body? Which area of the water body was estimated or quantified? Same for the riparian zone: which distance from the shore was considered riparian, and how these percentages were estimated/quantified?

You could highlight the dominant type of riparian vegetation to help readers focus.

Benthic invertebrates were sampled using hand netting…
meaning that samples were composed by combining all habitats?

126 how were feeding groups distinguished in the field? Based on morphology?

how is dispatch with boiling water humanely? To what end were they kept alive for 24 hours? And why did you use this boiling water method, instead of preservatives like alcohol or formol. This is related to the isotopic signal, but the reader does not know this.
Explain why their shells were remove.
I don’t see how conductivity and pH can affect the food web structure or isotopic niche. Oxygen and temperature, indirectly, if they reflect productivity?
Why was toxicity for macrophytes and algae estimate?
So far, FFGs have not being defined or described. What are they, what do they represent, how do you assign an organism to a FFG, or which categories exist. This table should be probably moved to the Results.
family determination based on which literature? FFGs can vary in different regions because organisms are flexible in their diets, even given certain morphology. Moreover, they can usually be assigned to more than one category. Even in the book of Merrit, Cummins, and Berg (the “inventors” of FFGs), taxa are usually categorized in more than one group.

Table 3. CPOM and FPOM are in the legend but not the table. Columns after the TU are not explained. How were all these metrics quantified and to what aim? Move to Results.

was observed
of all invertebrates or for primary consumers? This does not seem to be an appropriate justification, from my point of view, precisely because you are looking for potential differences between water body types. To my knowledge, there are two ways of correcting dN: first, by using a basal resource dN value as a baseline; this might have the problem of having different values among sites, if they differ in the amount of fertilizer. Second, using the dN value of the primary consumer with the lowest dN to obtain comparable trophic positions across sites.
difference between what and what?
what difference?
an alternative is centering or scaling the deltas of all sites to obtain comparable values, but I don’t understand why to alter the variability of the sites showing effects of agriculture, if that’s your question? Moreover, it seems odd and a bit arbitrary to transform values of one site.
Explain what these analyses are and why were they performed
what information
to what end? What is the information provided by these metrics?
I don’t understand why do you want to correct for the effects of agriculture? Also, is this another correction or the same one mentioned in the previous subsections?

Results

265 it hasn’t been explained or justified why these two types of habitats would differ in the isotopic composition

Figure 4 the resolution of the figure is very poor, please improve it.

replace clearly by a more appropriate term, like statistically
… for the MNND (Fig 5c)…
evenness is Fig 5f, not e
Which clear differences, between what and what? Please, try to avoid using words like “clear” or “clearly” since patterns might not be as clear or obvious for the readers.

Figure 6: the boxes correspond to SEA.B or SEAc?

284 this paragraph needs to be more organized. It is mixing the isotope values with abundances and diversity for one FFG.

306-309 these sentences are quite hard to read; please, revise the grammar.

were the explanatory variables scaled before running the analysis?

Discussion

but how was this tested?
why did you expect more abundance and diversity in ponds?
was any of this measured? Is the relative abundance of herbivores significatively higher in ditches? This all seems very speculative
homogenous in which sense? How was homogeneity measured?
then, why citing 46?
I don’t think secondary production is an appropriate term here, since this was not measured. Secondary production is a complex metric requiring, at the very least, biomass estimation.
again, secondary production was not measured.
this paragraph is generally confusing. I’d recommend you explain your results at the beginning, as a topic sentence, and then develop the ideas that support your findings. In the current version, sentences supporting and contradicting your findings are mixed making it hard for readers to get the main message you want to transmit.
is there a better term than indifferent? I think generalist could be more appropriate
meaning enriched values either due to high levels of omnivory or secondary predators.
herbivorous feeding preference
but you measured pollution. This is something you should be able to test.
why is this related to wider habitats?
lower diversity of littoral microhabitats in ditches? You started your paragraph talking about ponds.
a well-documented effect of eutrophication is the development of cyanobacteria, that outcompete other primary producers, but are commonly toxic for animals, explaining why eutrophication could actually reduce grazers diversity.
it is not clear what do you mean by wider habitat? Is the width of the water body, or the number of microhabitats available?

374 higher taxa richness where? These results were not presented before

375 what is a taxa group?

381 could pesticides have important direct effects on the invertebrates, leading to higher mortality rates, in addition to the indirect effects through primary production?

386 it makes sense for me to have higher variability in these groups related to their diets; they are after all generalists

393 Why?

405 I am surprised to read here that ponds were supposed to be a reference community. This kind of statement should be mentioned very early in the Methods or Introduction. Moreover, I don’t think ponds embedded in an agricultural landscape could be used as a reference, less impacted community by agriculture. I’d suggest using the water bodies as a gradient of agricultural effects, including nutrient enrichment, pesticides, and the proportion of surrounding crops to test your hypotheses.

Reviewer #2: Manuscript:

Assessing agricultural effects on benthic invertebrate communities in ponds and ditches

using δ¹⁵N and δ¹³C isotope niches

Authors: Fee Nanett Trau, Kathrin Fisch, Stefan Lorenz

Overall, the manuscript presents interesting and valuable research, but several key areas require further clarification to improve its structure, transparency and readability. The study's aim is currently too vague and should be made more specific, particularly in regard to which aspects of benthic invertebrates are being investigated. The introduction would benefit from a clearer distinction between ponds and ditches, specifying their purpose, shape, and typical formation or usage.

In the methods section, there is a lack of clarity surrounding the toxicity calculations mentioned. It is important to explicitly connect these methods to the study's objectives so the reader can follow the logic of your approach. At the moment, the description of toxicity tests, seem disconnected from the rest of the paper and need to be more clearly integrated into the study's goals. If toxicity testing was conducted, please describe the specific methods used, including whether test organisms or particular pesticides were involved. Further, the process for sorting organisms into taxonomic and functional feeding groups (FFGs) should be described more clearly. The use of boiling water as a method needs more explanation- if this method is employed instead of freeze-drying, please justify why it was chosen and explain how it might affect the isotopic analysis.

It would be valuable to emphasize whether the observed differences between ponds and ditches are statistically significant. Discussing the statistical significance of these differences would provide a clearer understanding of their reliability and help contextualize the results within the broader field. Additionally, providing information about the statistical tests used, the p-values, and any relevant confidence intervals could strengthen the interpretation of the findings and give readers more confidence in the conclusions drawn.

Some aspects of the figures and tables require additional detail. For example, Figure 1 should have a more informative caption, specifying what the red labels represent and distinguishing between the different symbols. Some Tables also need clearer explanations.

In addition, there are a number of minor issues that need addressing. It would be useful to include more details on the taxonomic composition of riparian vegetation, ideally at the family level. The citation format should be checked according to the authors' guidelines to ensure consistency. Also, please specify the number of replicates used at each location and for each taxonomic or functional feeding group, as well as the amount of material weighed for each replicate sample.

More specific comments and suggestions are provided in the attached PDF. Addressing these points will strengthen the manuscript, making it clearer, more precise, and easier for readers to follow.

**Do you want your identity to be public for this peer review?** For information about this choice, including consent withdrawal, please see our Privacy Policy

Reviewer #1: **Yes: ** Daniela Cortes Guzman

Reviewer #2: No

---

## [Author Response · Author response to Decision Letter 1]

12 May 2025

Author’s response:

We thank the Academic Editor for handling our manuscript and for the careful selection of qualified Reviewers. We are glad to receive such a constructive, detailed, and fair review report. We revised our manuscript according to the comments of the Reviewers.

Comments to the Author

1. Is the manuscript technically sound, and do the data support the conclusions?

Reviewer #1: Partly

Reviewer #2: Partly

2. Has the statistical analysis been performed appropriately and rigorously?

Reviewer #1: I Don't Know

Reviewer #2: Yes

3. Have the authors made all data underlying the findings in their manuscript fully available?

Reviewer #1: Yes

Reviewer #2: Yes

4. Is the manuscript presented in an intelligible fashion and written in standard English?

Reviewer #1: Yes

Reviewer #2: Yes

5. Review Comments to the Author

6. PLOS authors have the option to publish the peer review history of their article (what does this mean?). If published, this will include your full peer review and any attached files.

Do you want your identity to be public for this peer review? For information about this choice, including consent withdrawal, please see our Privacy Policy.

Reviewer #1: Yes: Daniela Cortes Guzman

Reviewer #2: No

Reviewer #1:

I finalized revising the manuscript “Assessing agricultural effects on benthic invertebrate communities in ponds and ditches using δ¹⁵N and δ¹³C isotope niches”. The topic is interesting and attractive to a broad audience. However, major improvements need to be done in the manuscript. The questions and hypotheses, and how the methods were applied to solve these questions, were rather cryptic for me. It was generally hard to understand from the introduction which is the knowledge gap the authors are trying to fulfill. Moreover, without clear questions, assessing the science behind the methods and results was difficult. My major misunderstanding is why the authors expect differences between the two types of ecosystems assessed, and therefore, why this is an interesting question in the framework of agricultural stressors. I think the authors have an interesting and valuable dataset with the potential of answering interesting ecological questions. However, significant work needs to be done. I left detailed comments for each of the sections.

Author’s response:

We thank the reviewer for the time that she invested in evaluating our manuscript and we hope that we have addressed all the concerns raised by the reviewer accordingly.

Comments reviewer 1 with authors responses:

Abstract

14: Why do ponds and ditches harbor particularly endangered or scarce taxa? Moreover, why do you mean by scarce, is it low abundant taxa or taxa with low occupancy? I’d recommend removing this last part of the sentence.

Explained in more detail why these water bodies are valuable habitats and replaced scarce with rare.

16 I disagree. Numerous studies have shown the adverse effects of agricultural land use in biodiversity. Perhaps what you mean is that these habitats are likely to face multiple stressors, all derived from agriculture, which results in a problematic complex to address. More importantly, the reader does not know so far which are these previous approaches and why they were not successful. For example, what do you mean by “classic structural biodiversity metrics”?

Added the word “indirect or multiple” to stressors and examples for the classical metrics mentioned. Problems with previous studies will be elaborated on in the introduction section. The sentence in the abstract should only function as a teaser for the reader.

19: We analyzed the carbon and nitrogen stable isotope composition of the invertebrates and the pesticides and nutrient residues of water.

Addressed.

21 readers do not know what are Standard ellipse and Layman B community metrics. I recommend leaving technical terms out of the abstract.

Addressed. Terms left out and circumscribed as “Estimates of community metrics…”

22 FFG should be introduced earlier.

FFGs are introduced earlier in the abstract when the general approach is explained. Line 18 ff. “Benthic invertebrates, from six different functional feeding groups (FFGs), and water samples were collected from three ponds and four ditches in an agricultural landscape in Brandenburg, Germany.”

Which differences? You’re leaving the reader hanging on the unexpected differences but don’t tell them (us) which.

Explanation for predator example added. “…, which was more abundant and diverse in ditches.”

28-32 See my previous comment. These sentences are uninformative for readers. We don’t know why case-by-case patterns emerged, or where the differences lay. What do you mean by the production or consumption level? How was production measured or estimated? We also don’t know which strong fluctuations on parameters did you observed; therefore, we don’t know how did you reach this conclusion.

Addressed. Deleted the sentence from the abstract as it is confusing here and will be elaborated in the results and discussion.

Remove In the end

Addressed.

Introduction

is it specialization, and use of fertilizer and pesticides homogenizing the landscape? I’d say that the expansion of agricultural land is what homogenizes the landscape. Ref 3

I would still argue that it is specialisation and the increased use of inputs in intensive or industrial agriculture that makes a landscape more homogenous as intensive agriculture usually goes hand in hand with larger plots being used for one crop, which can be sprayed and harvested by large machines. The expansion of agriculture is another point that can also lead to homogenization when land is converted to intensive agricultural land. Thus, I kept the sentence but added expansion to the explanations in the sentence.

What do you mean by semi-natural habitats? Under less intensive agriculture?

I mean habitats such as ponds, ditches, field margins or hedges. In general, habitats that were altered by humans but still have a high biodiversity value. Examples added in text.

Starting a paragraph with an example is quite counterintuitive. You haven’t introduced yet non-productive areas, so why are you giving us examples?

I connected the paragraph with the paragraph before, as this is the explanation with examples for the semi-natural areas. See also comment above.

Same as in the abstract: would you define scarce more precisely.

Replaced scarce with rare.

a high richness compared to what? Also, these groups are highly abundant in many aquatic ecosystems; therefore, why are they particularly important in ditches?

I changed the sentence to make my intention with that sentence clear. “Also ditches, which are in close proximity to agricultural practices, have the potential to host diverse communities as well as rare or endangered aquatic species (9) and show a high richness for the groups Coleoptera, Gastropoda, Trichoptera and Heteroptera (9).

I think this is your topic sentence for this paragraph; therefore, it should be the first one -or maybe the second one, but at the beginning!

Paragraph reordered:

“Ponds and ditches are biodiversity hotspots and form aquatic corridors in a landscape dominated by agriculture in need of conservation action (5,6). Unfortunately, these small water are usually not included in monitoring programs (5,7), but the importance of particularly ponds, but also ditches, for benthic macroinvertebrate biodiversity and ecosystem services is increasingly gaining recognition in research activities (5,9–14). Ponds are known to harbour high species numbers (9), however also ditches, which are usually in close proximity to agricultural practices, have the potential to host diverse communities as well as rare or endangered aquatic species (5) and show a high richness for the groups Coleoptera, Gastropoda, Trichoptera and Heteroptera (5).”

but this contradicts your previous sentence in line 48

Yes true, deleted the contradicting part. The main point here was that those small water bodies are not included in monitoring programs.

in which sense is challenging? Actually, invertebrates are one of the most used groups for biodiversity assessment because sampling is comparatively easy and cheap to other taxa, and they are very good indicators of ecosystem status and functions.

Added explanation in the sentence. “…the assessment of water or habitat quality by benthic invertebrate biodiversity is challenging…”

But there is extensive literature showing the effects of agriculture on biodiversity. I think you need to be more specific as to why functional approaches outperform, or complement, taxonomic approaches. For example, why would functional approaches be more sensible to persistent stressors or short gradients? Ref 8

Added: “Additionally species occurrences may be subject to some degree of stochasticity (15) and functional responses can reflect the ecological conditions and habitat change better than taxonomic approaches (16,17).”

why were these other traits not included in your analysis? One important effect of agricultural land use is the change in inputs of organic matter to the water bodies; this justifies the use of feeding groups. However, other effects are channelization or modification of the physical structure, which might produce a response in locomotion; changes in productivity (eutrophication), with effects on body size and respiration; changes in temperature, with effects on thermal tolerance and respiration. In other words, you need either to include more traits, or to justify why feeding groups are the best approach to address the effects of agriculture, including persistent stressors and short gradients.

Channelization and physical modification are important factors for (natural) streams (or lakes for physical modification) suffering from anthropogenic pressure. Ditches are per se longitudinal channels, all of which are constantly modified in their physical structure by bank cleaning and mowing. Therefore, we assume communities to be adapted to this environment.

Added in the manuscript: “We selected FFG to address the effects of agriculture as further stressors such as channelization and modification of physical structure do not vary in their intensity in the selected water bodies”

this paragraph is disconnected with the rest of the introduction. While important concepts need to be defined, the reader wants to know how the niche concept(s) is related to your own work and very importantly, to your question and knowledge gap. It is your task, as writers, to tell the reader how this concept can help understanding effects of stressors in biodiversity -not give the reader homework . For example, I’d recommend combining this paragraph with the trait paragraph, since the niche is going to be defined based on those traits. This will provide more coherence between your paragraphs and shorten your introduction.

Addressed. Combined with the trait paragraph as suggested. This also makes clear that the adaptation of the niche concept in stable isotope studies is done to create isotopic niches, which are a proxy for the habitat and resource use of a community. Habitat and resources are linked to the land use and agricultural stressors. This link was stressed in the text.

we used. You haven’t defined isotopic niche nor realized niche. Feeding resources? And which habitats? Do you mean between ponds and ditches? If so, why feeding resources and, therefore, isotopic niches would differ between them?

Described in more detail above in the manuscript.

we compared

Addressed

which groups?

Addressed. “…compared niche widths of FFGs and the pond and ditch communities based on…”

are you assessing more than one stressor? Or is it pesticides the only metric you are using?

No, pesticides and nutrients are examined. Changed in the text as follows: “…influence of agricultural stressors (toxicity of water samples due to pesticide residues and eutrophication due to nutrient inputs)…”

we asked. Can

Adressed

I don’t fully understand your questions. Are you assessing isotopic niches in a gradient of agricultural stress, as measured by pesticides? Why do you expect differences, or not, between ponds and ditches?

Added in the text: “We assume that ponds and ditches are influenced differently by agricultural stressors. Ditches are expected to be more strongly influenced by agricultural stressors than ponds due to their characteristics, which are a rather homogeneous habitat in and around the water body, a directly farmed environment with the location in the field (grassland) without a riparian buffer. “

Questions adapted: “We asked the following questions, using ponds and ditches as an example.: Can the isotopic niches depict the effects of agriculture on benthic invertebrate communities in the selected water bodies in the study region? Are the observed patterns of isotopic niches of FFGs comparable in ponds and ditches for the FFGs?”

to answer these questions…

Adressed

what is the difference between niche differentiation and niche specialization? And why reduced niche specialization would result in less overlap? I’d expect the opposite. How about the expectations for ponds? Do ponds have different effects of agriculture?

The term differentiation and specialisation mean the same. Niche differentiation = competing species use the niche in different ways to coexist, different use of resources, portioning of space and time. The word specialisation was replaced with differentiation in the second part of the sentence.

Yes, it should be “larger overlaps”. Changed in the text.

It is not clear to me which questions are being answered and how. Is it the question related to differences in niche width due to different levels of agricultural stress? Then, would these effects differ between ponds and ditches? Why? What do you mean by habitat variability? Is this variability inside each water body, or among types?

Added in the text, see above.

Added text at the beginning of the paragraph: “….affecting habitat availability and quality and thus influencing benthic invertebrate food webs and niche size (33,34).”

Added text: “Ditches, for example, are probably more contaminated, but as they are part of a flow system, pollutants can also be transported downstream. Ponds, on the other hand, are better protected from pesticide inputs, but can act as a source and sink for pesticide residues in the sediment (35).”

Methods

I thin

---

## [Decision Letter · Decision Letter 1]

28 May 2025

Dear Trau,

Thank you for submitting your manuscript to PLOS ONE. After careful consideration, we feel that it has merit but does not fully meet PLOS ONE’s publication criteria as it currently stands. Therefore, we invite you to submit a revised version of the manuscript that addresses the points raised during the review process.

**ACADEMIC EDITOR: **

Dear authors, many thanks for revising your manuscript. Two referees have evaluated it again and despite some improvements were made, the manuscript still needs significant work before its consideration for publication, particularly in terms of writing and interpretation. I invite you to go in detail through both reviewers’ comments and suggestions and modify the manuscript accordingly.

We look forward to receiving your revised manuscript.

Kind regards,

Marina Vilenica

Academic Editor

PLOS ONE

Reviewers' comments:

Reviewer's Responses to Questions

**Comments to the Author**

Reviewer #1: (No Response)

Reviewer #2: (No Response)

2. Is the manuscript technically sound, and do the data support the conclusions?

Reviewer #1: Partly

Reviewer #2: Yes

3. Has the statistical analysis been performed appropriately and rigorously?

Reviewer #1: No

Reviewer #2: Yes

4. Have the authors made all data underlying the findings in their manuscript fully available?

Reviewer #1: Yes

Reviewer #2: Yes

5. Is the manuscript presented in an intelligible fashion and written in standard English?

Reviewer #1: No

Reviewer #2: Yes

Reviewer #1: The manuscript has improved. The authors addressed some of the concerns; however, considerable work is still needed in order to have a manuscript ready for publication. It seems to me that some changes in the manuscript were made to please the reviewers, but without giving deeper thought about why these requests were made. In that sense, changes lack attention to detail. For example, Table 3 was moved to results, as suggested, but it is not described, and Figure 4 is missing. Furthermore, I disagree with the interpretation of results the authors made. They found statistical differences in several metrics yet concluded that there are no clear patterns.

I have several major and minor comments across all sections that I detailed here:

Abstract

15: the last part of the sentence, after landscape is a bit redundant and unnecessary; it could be removed.

15-20 I still find the wording here difficult to justify the use of functional approaches (or not use of taxonomic ones). Might I suggest a phrasing like: Biodiversity metrics based on taxonomy, such as richness or evenness indices, might fail to describe changes in ecosystem functions that result from the effects of agriculture. In contrast, functional approaches, such as the use of functional feeding groups and isotopic composition, better describe the use of resources by the invertebrates, improving our understanding of the influence of agricultural stressors on aquatic communities (just a suggestion, please use your own words)

we still don’t know why these differences are unexpected. I suggest removing that word.

33-34 this seems to be part of your results, rather than a general statement. I would write it in past tense. More importantly, this might be, in effect, a consequence of your small number of water bodies. Therefore, I don’t think your sentence at line 31 is justified: communities respond in complex ways to stressors… since this is, again, an effect of the sample size. I suggest re-ordering this last section of the abstract to provide more accurate conclusions. For example: The strong fluctuations in biotic and abiotic parameters resulted in pronounced differences among water bodies, but no clear patterns between water body types. However, larger datasets… However, after reading your main text, I disagree with this conclusion.

Introduction

water bodies. Suggested phrasing: Although these small water bodies are usually not included in monitoring programs (5,7), their importance for benthic macroinvertebrate biodiversity and ecosystem services is increasingly gaining recognition in research activities (5,8–13).

53-56 These last sentences are a bit repetitive. It was already stated that these ecosystems are valuable habitats for invertebrates. Moreover, the sentence structure is a bit confusing. I think mentioning the orders at the end of the sentence is unnecessary since, as I have suggested before, these are abundant and common groups in many aquatic ecosystems. My previous comment was not meant to have an aesthetic change in the wording, but to justify the inclusion of such a sentence in your introduction. I recommend closing the paragraph with a statement highlighting the knowledge gap that the manuscript is addressing.

58-59 I still disagree on that it is challenging being one of the most common bioindicators. I think you could instead jump directly into the previous findings and avoid this sentence.

62-63 long-term persistent stressors would be likely reflected in taxonomic diversity or EPT, whereas short stressor gradients might result in stochasticity, for which functional approaches consistently reflecting habitat types and resources might be a solution

the acronym FFG has not been defined in the introduction. Please do so

69-71 this sentence does not belong here since you haven’t told us yet anything about the water bodies under study. The justification for the use of FFG, from an ecological point of view, is your next sentence.

explain why they reflect the effects of agriculture. Is it because of the change in quality, quantity, type of feeding resources?

74-78 this definition of niche could be better described in terms of the feeding/trophic niche, given that it is the topic of the manuscript and to better connect with the previous section of the paragraph.

please define realized ecological niche
previous sentence was spelled realized... please, confirm with the journal British or American spelling

93-94 not necessary if the niche is explained in that context in the previous sentences

95-98 this paragraph could be written more concisely. Suggestion: We aimed to assess the influence of agricultural stressors, measured as water toxicity from pesticide residues and eutrophication, on the isotopic niche of FFGs in pond and ditch communities.

106-108 this phrasing is confusing. In my understanding, is not the body type, pond or ditch, what is used as a gradient (since this a categorical factor with only two levels; if so, which is higher and which lower?). It is the continuous values of nutrient enrichment, pesticides, and the proportion of surrounding crops.

I am not sure I understand your second question. Do you mean that, on one side, you expect differing effect of agriculture between ditches and ponds on the food web structure (Q1 and H1), but similarities in the isotopic niches of specific FFGs (Q2 and H?)?
... and presence of generalists compared to ponds. To keep parallelism and include the ponds in your hypothesis.

Methods

and ditch 4?
specimens
otherwise? You collected as many as possible or discarded samples that did not reach 10 individuals?

159, 161 grounded

10-15 samples, even if they belonged to different genera?
isotopic... were carried
that can explain... I still don’t understand this. Why do you expect that conductivity and pH explain the food web structure? Your answer was unfortunately not sufficient. I know these are standard variables typically measured during sampling, but that does not justify their inclusion in your statistical analysis. I could understand that you included them to control for intrinsic differences among water bodies; therefore, they are not explanatory but control variables.
explain why these metrics were calculated, or what information did you get from them.
check parallelism of the sentence
why do you have small samples sizes in EC and oxygen if they were measured with a probe?

Results

You have to describe the table, not just reference it. For example, what common patterns did you observe across ponds, and in comparison to ditches? Which type of water body tended to have higher nutrient concentrations, toxicity, or pesticides?

Table 3. For the sake of completeness, describe all acronyms. Sample was used before to describe the individuals analyzed for isotopic composition. I recommend using here Site or water body. C° are the units of temperature. To keep it consistent with your other columns, use T [°C] or something similar.

Fig 4 is missing. Is it maybe Fig S2? If so, could you show the biplot after correction? Maybe this will distinguish

dC13 range is 5d not b
whereas
is this Figure S3? What different information is provided between Fig 5 and S3? It seems that many of the differences shown in Fig 5 disappeared in S3. This should be important to explain. In a more general comment, all tables and figures in the text are meant to be described and explained, not just mentioned.
since Fig. 6 is comparing FFG among wb types, I suggest placing each FFG in pond and ditches next to each other, and distinguish them with colors, as you have done. This way is easier for the reader to notice the differences between, e.g., predators between water body types. You also need less labels in the x axis this way.
I disagree with your initial statement. I think there are differences between ponds and ditches, as you mentioned in the following sentences.
what about the diversity of the other groups? Also, you didn’t really quantify abundance, since you stopped gathering individuals when you collected 10.
for all FFG, overlap was greater when corrected for both C and N. Moreover, why this comparison between corrections is important? Is one correction providing different information than the other? Why not simply correct by both and compare overlaps?
a lower ... in ponds compared to ditches, and higher... in ditches...
higher
this is the description of table 3 that should be with the table, not in a different section.

Discussion

what exactly do you mean by comparable? The isotopic niche width? However, you found marked differences in Fig 5
I don’t agree with this interpretation. Again, you found statistical differences in some niche metrics and in some FFGs.
since the isotopic niche is indirectly measuring feeding resources, I’d recommend sticking to that word and not use it together with habitats.
what patterns did you expect to find? Since your main message seems to be that there were no differences among habitats, despite several metrics in your results showing differences, I wonder which were the patterns you expected to observe?
however, in your introduction you stated that ditches could also harbour high diversity. Therefore, why is this contrary to your expectations?
but is this true? Did you observe more plants or algae in ditches? Did you observe higher diversity of primary consumers? Moreover, this is assuming that the food web is based on primary production, but disregarding the role of detritus to sustain aquatic food webs, this is, all allochthonous inputs.
but you didn’t compare species richness
what do you mean by more homogeneous communities for predators? This term could be confusing, and it is not to be compared to having lower diversity.
this paragraph is describing the results from the figure, not discussing the further implications of such results. It should be moved to the Results section and discussed in the Discussion. Moreover, I failed to see why the ample preference of omnivorous is restricted to polluted waters, since they are, by definition, feeding on multiple types of resources.
same, these are Results. Please reference the figure or table that they refer to.
as mentioned before, I recommend you keep your interpretation in terms of the feeding resources, not the habitats or microhabitats
but you found higher diversity in ponds. Where is the link with the lake’s reference?
what do you mean by primary producer-rich? As in having higher primary production? Measured how? Or as in having higher diversity of primary producers? Measured how?
see my previous comments on this conclusion

436-439 these sentences are disconnected to the starting sentence of the paragraph.

explain why their food resources would be low in November
in deed, organisms need current to filter; therefore, it is expected to have low densities of filterers in water bodies without current.
explain why sampling in autumn affects the detectability of invertebrates and pesticides
explain why is this recommended

Conclusions

If the other sections are revised, this section would also require modifications accordingly.

Reviewer #2: Dear Authors,

Thank you for your thorough revision of the manuscript. It is evident that you have made significant efforts to address the reviewers' comments, and many aspects of the manuscript have improved as a result. That said, there are still several points that require further attention to enhance the clarity, coherence, and scientific rigor of the work.

Most notably, improvements to the language are recommended to increase readability and avoid unnecessary or unclear repetitions. In particular:

_Some sentences are redundant, as the information they convey has already been clearly stated in preceding text.

_Paragraph structure can be improved by shifting certain sentences to new paragraphs for better flow and thematic clarity.

_Several sentences would benefit from rephrasing to improve clarity and avoid grammatical issues such as the repetitive use of conjunctions (e.g., multiple occurrences of “but” in the same sentence).

_In a few instances, sentences are overly long and difficult to follow, and should be restructured into more concise and digestible forms.

_All abbreviations should be defined at their first mention in the text to ensure clarity for all readers.

_Certain terminology (e.g., "realized niche") may be unclear in this context. More precise phrasing is suggested to better convey the intended meaning.

_Some key methodological choices would benefit from the inclusion of appropriate references to justify their use.

_As a general comment, enhancing the narrative structure and clarity of expression will improve the overall impact of the manuscript and make it more accessible to a broader readership. A careful final language edit is strongly recommended.

For more detailed suggestions and specific examples, please refer to the attached PDF version of the comments.

**Do you want your identity to be public for this peer review?** For information about this choice, including consent withdrawal, please see our Privacy Policy

Reviewer #1: **Yes: ** Daniela Cortes

Reviewer #2: No

---

## [Author Response · Author response to Decision Letter 2]

24 Jun 2025

Author’s response:

We thank the Academic Editor for handling our manuscript and for the careful selection of qualified Reviewers. We are glad to receive such a constructive, detailed, and fair review report. We revised our manuscript again according to the comments of the Reviewers. We hope to address all raised questions and issues made by the reviewers. Please go through the following comments and authors response.

Comments to the Author

1. If the authors have adequately addressed your comments raised in a previous round of review and you feel that this manuscript is now acceptable for publication, you may indicate that here to bypass the “Comments to the Author” section, enter your conflict of interest statement in the “Confidential to Editor” section, and submit your "Accept" recommendation.

Reviewer #1: (No Response)

Reviewer #2: (No Response)

2. Is the manuscript technically sound, and do the data support the conclusions?

Reviewer #1: Partly

Reviewer #2: Yes

3. Has the statistical analysis been performed appropriately and rigorously?

Reviewer #1: No

Reviewer #2: Yes

4. Have the authors made all data underlying the findings in their manuscript fully available?

Reviewer #1: Yes

Reviewer #2: Yes

5. Is the manuscript presented in an intelligible fashion and written in standard English?

Reviewer #1: No

Reviewer #2: Yes

7. PLOS authors have the option to publish the peer review history of their article (what does this mean?). If published, this will include your full peer review and any attached files.

Do you want your identity to be public for this peer review? For information about this choice, including consent withdrawal, please see our Privacy Policy.

Reviewer #1: Yes: Daniela Cortes

Reviewer #2: No

Reviewer #1

Review Comments to the Author

The manuscript has improved. The authors addressed some of the concerns; however, considerable work is still needed in order to have a manuscript ready for publication. It seems to me that some changes in the manuscript were made to please the reviewers, but without giving deeper thought about why these requests were made. In that sense, changes lack attention to detail. For example, Table 3 was moved to results, as suggested, but it is not described, and Figure 4 is missing. Furthermore, I disagree with the interpretation of results the authors made. They found statistical differences in several metrics yet concluded that there are no clear patterns.

I have several major and minor comments across all sections that I detailed here:

Author’s response:

We thank the reviewer for the time that she invested in evaluating our manuscript again and we hope that we have addressed all the concerns raised by the reviewer now.

Comments reviewer 1 (black) with authors responses (green):

Abstract

15: the last part of the sentence, after landscape is a bit redundant and unnecessary; it could be removed.

Addressed, part removed.

15-20 I still find the wording here difficult to justify the use of functional approaches (or not use of taxonomic ones). Might I suggest a phrasing like: Biodiversity metrics based on taxonomy, such as richness or evenness indices, might fail to describe changes in ecosystem functions that result from the effects of agriculture. In contrast, functional approaches, such as the use of functional feeding groups and isotopic composition, better describe the use of resources by the invertebrates, improving our understanding of the influence of agricultural stressors on aquatic communities (just a suggestion, please use your own words)

Addressed as follows: “Due to a lack of sensitivity biodiversity metrics based on taxonomy only, such as overall taxa number, Shannon or evenness index, might fail to detect changes in communities, which lead to changes in ecosystem functions resulting from agricultural practices. In contrast, functional approaches, such as the use of functional feeding groups and isotopic composition, better describe the utilisation of resources by invertebrates and improve our understanding of the impact of agricultural stressors on benthic invertebrate communities in ponds and ditches.”

29 we still don’t know why these differences are unexpected. I suggest removing that word.

Addressed. “unexpected” was deleted

33-34 this seems to be part of your results, rather than a general statement. I would write it in past tense. More importantly, this might be, in effect, a consequence of your small number of water bodies. Therefore, I don’t think your sentence at line 31 is justified: communities respond in complex ways to stressors… since this is, again, an effect of the sample size. I suggest re-ordering this last section of the abstract to provide more accurate conclusions. For example: The strong fluctuations in biotic and abiotic parameters resulted in pronounced differences among water bodies, but no clear patterns between water body types. However, larger datasets… However, after reading your main text, I disagree with this conclusion.

Statement written in past tense and text changed as follows: „The strong fluctuations in biotic and abiotic parameters led to pronounced differences between the water bodies. Our study showed that benthic invertebrate communities in small unstable systems respond in complex ways to stressors.

Introduction

51 water bodies. Suggested phrasing: Although these small water bodies are usually not included in monitoring programs (5,7), their importance for benthic macroinvertebrate biodiversity and ecosystem services is increasingly gaining recognition in research activities (5,8–13).

Changed as suggested.

53-56 These last sentences are a bit repetitive. It was already stated that these ecosystems are valuable habitats for invertebrates. Moreover, the sentence structure is a bit confusing. I think mentioning the orders at the end of the sentence is unnecessary since, as I have suggested before, these are abundant and common groups in many aquatic ecosystems. My previous comment was not meant to have an aesthetic change in the wording, but to justify the inclusion of such a sentence in your introduction. I recommend closing the paragraph with a statement highlighting the knowledge gap that the manuscript is addressing.

Addresses, repetitive sentences deleted. Paragraph connected with the first sentences of the following paragraph and statement highlighting knowledge gap added as follows: “…This study will contribute to gain knowledge on measuring changes in communities, which is indispensable in order to conserve and promote these aquatic habitats and their biodiversity. Thorough knowledge on the status and indicators for measuring biodiversity is needed.”

58-59 I still disagree on that it is challenging being one of the most common bioindicators. I think you could instead jump directly into the previous findings and avoid this sentence.

Addressed. Deleted as suggested.

62-63 long-term persistent stressors would be likely reflected in taxonomic diversity or EPT, whereas short stressor gradients might result in stochasticity, for which functional approaches consistently reflecting habitat types and resources might be a solution

I partly agree. Long-term persistent stressors can be reflected in taxonomic diversity and EPT only when taxonomic reference data exists. Usually there is no historical data available from communities that were not exposed to pesticides etc.

69 the acronym FFG has not been defined in the introduction. Please do so

Addressed.

69-71 this sentence does not belong here since you haven’t told us yet anything about the water bodies under study. The justification for the use of FFG, from an ecological point of view, is your next sentence.

Agreed. Sentence deleted. The sentence that follows in the manuscript is the ecological explanation.

73 explain why they reflect the effects of agriculture. Is it because of the change in quality, quantity, type of feeding resources?

All the changes you mentioned are important and can be reflected. Made two sentences out of the part and revised the second sentence as follows: “Such analyses also provide valuable insights into the impacts of agricultural activity, including habitat fragmentation, degradation, and pollution, which alter resource quality, quantity, and type, ultimately influencing community structure (17,22–24).

74-78 this definition of niche could be better described in terms of the feeding/trophic niche, given that it is the topic of the manuscript and to better connect with the previous section of the paragraph.

Sentence to better connect the two paragraphs added: “Understanding how species interact with their environment and with each other is therefore essential for interpreting changes in community structure. In this context, the niche concept provides a framework for linking species traits and resource use with ecosystems.”

Further explanation and descriptions of the isotopic niche follow in the paragraph later. I started with the general niche concept and then went into stable isotopes and their meaning and then to the term isotopic niche. Otherwise the reader does not know what I mean with isotopic I think.

91 please define realized ecological niche

Addressed as follows: “The isotopic niche can thus serve as a proxy for the realized ecological niche of a species (portion of the ecological niche that a species occupies in nature, after accounting for competition and predation (31)) reflecting their use of resources and habitat (32). The realised niches are linked to land use and agricultural stressors, which alter resource availability and habitat quality.”

92 previous sentence was spelled realized... please, confirm with the journal British or American spelling

Under the term language they just state “Manuscripts must be submitted in English.” (https://journals.plos.org/plosone/s/submission-guidelines). I changed it to the American English spelling.

93-94 not necessary if the niche is explained in that context in the previous sentences

I would rather keep it to make it clear which niche term I am using to the reader. Especially as I am talking about the realized niche and the isotopic niche in the sentence before.

95-98 this paragraph could be written more concisely. Suggestion: We aimed to assess the influence of agricultural stressors, measured as water toxicity from pesticide residues and eutrophication, on the isotopic niche of FFGs in pond and ditch communities.

I followed your suggestion and kept the last part of my sentence to mention also the method: “We aimed to assess the influence of agricultural stressors, measured as water toxicity from pesticide residues and eutrophication, on the isotopic niche of FFGs in pond and ditch communities, using a geometric approach originally developed by (24).”

106-108 this phrasing is confusing. In my understanding, is not the body type, pond or ditch, what is used as a gradient (since this a categorical factor with only two levels; if so, which is higher and which lower?). It is the continuous values of nutrient enrichment, pesticides, and the proportion of surrounding crops.

I rephrased the sentence as follows: “Ponds and ditches representing water body types with varying exposure to agricultural effects (nutrient enrichment, pesticides, and the proportion of surrounding crops) were used to test our hypotheses.”

I am not sure I understand your second question. Do you mean that, on one side, you expect differing effect of agriculture between ditches and ponds on the food web structure (Q1 and H1), but similarities in the isotopic niches of specific FFGs (Q2 and H?)?

For your explanation. Question one is asking if we can depict the effects of agriculture on benthic invertebrate communities with the isotopic niches, can we see a gradient for example. In the text: “Can the isotopic niches depict the effects of agriculture on benthic invertebrate communities in the selected water bodies in the study region?”

Question two is asking if we do see the same image/pattern in ponds and ditches. In the text: “Are the observed patterns of isotopic niches of FFGs comparable in ponds and ditches for the FFGs?”

... and presence of generalists compared to ponds. To keep parallelism and include the ponds in your hypothesis.

Added as suggested: “On the other hand, we hypothesize that trophic structures in ponds are more differentiated and that we observe more specialized groups with less niche overlap.”

Methods

and ditch 4?

Addressed. It should be ditches 2 to 4.

specimens

Addressed.

otherwise? You collected as many as possible or discarded samples that did not reach 10 individuals?

Some more explanation was added in the text as follows: “Sampling was carried out until 10 specimens of the present orders/families were collected or as enough specimens were collected to create 10 pooled samples with the necessary sample weight as recommended by (30), whenever possible. If 10 specimen were not found a minimum of three samples was used for the analysis.”

159, 161 grounded

Addressed.

10-15 samples, even if they belonged to different genera?

No, samples were analysed for identified taxa separated. Identification was on family or genus level, if more than one genus was identified they were not mixed. Text changed as follows: “For each taxon identified, a maximum of 10 to 15 samples was analyzed.”

isotopic... were carried

Addressed.

that can explain... I still don’t understand this. Why do you expect that conductivity and pH explain the food web structure? Your answer was unfortunately not sufficient. I know these are standard variables typically measured during sampling, but that does not justify their inclusion in your statistical analysis. I could understand that you included them to control for intrinsic differences among water bodies; therefore, they are not explanatory but control variables.

The variables can explain differences between the water body types. Thus I would not name them control variables. Rephrased: “Environmental chemical parameters were measured to identify potential variables that could function to explain differences between the water body types.”

explain why these metrics were calculated, or what information did you get from them.

The explanation follows in the next paragraph. To make this clearer I added at the beginning of the paragraph “layman” before metrics in the text and deleted the word “described”.

269 check parallelism of the sentence

Rephrased to avoid parallelism and make sentences easier to folllow: “The SEA.c, based on SEA.B and adjusted for small sample sizes, provides and unbiased measure of isotopic niche width. (30). Larger SEA values indicate

---

## [Decision Letter · Decision Letter 2]

9 Jul 2025

Dear Dr. Trau,

Thank you for submitting your manuscript to PLOS ONE. After careful consideration, we feel that it has merit but does not fully meet PLOS ONE’s publication criteria as it currently stands. Therefore, we invite you to submit a revised version of the manuscript that addresses the points raised during the review process.

We look forward to receiving your revised manuscript.

Kind regards,

Marina Vilenica

Academic Editor

PLOS ONE

Reviewers' comments:

Reviewer's Responses to Questions

**Comments to the Author**

Reviewer #1: (No Response)

2. Is the manuscript technically sound, and do the data support the conclusions?

Reviewer #1: Partly

3. Has the statistical analysis been performed appropriately and rigorously?

Reviewer #1: No

4. Have the authors made all data underlying the findings in their manuscript fully available?

Reviewer #1: Yes

5. Is the manuscript presented in an intelligible fashion and written in standard English?

Reviewer #1: Yes

Reviewer #1: I have finished reviewing the manuscript for the third time. The manuscript has improved compared to the previous versions. However, I still have significant comments that, in my opinion, must be addressed before publication. I regret to say, once more, that the authors’ responses are occasionally not satisfactory. Moreover, while I appreciate their responses and explanations directed to me, my request and questions are finally made to improve understanding from the readers. In that sense, replying to me, but not fully addressing the issues in the text is not helpful for future readers of their work. Some explanations and changes made from the previous version to this one turned out to be more confusing to me. Specifically, the maize corrections for C. Some analyses and figures are presented with the correction, which leads to the conclusion that there are not differences between water body types, whereas others are presented without the correction, in which case significant differences emerged. In my previous comments, I asked about the different results when correcting versus not correcting, and the answer was that it was only intended as an additional visualization. However, the conclusions are dramatically different between the two methods.

Additional comments are detailed below.

Abstract

15 delete only, as they are not only based on taxonomy, they also account for abundance.

30 again, you didn’t measure abundance because you stopped counting when reaching 10 individuals. They were more diverse because you identified more taxa.

Introduction

54 for measuring biodiversity in ponds and ditches affected by agricultural activities… or something similar that provides context for the sentence.

86 taxonomic group since taxa is plural

87 for resource use or consumption

91 the habitat use is not reflected in isotopic compositions because it is not incorporated in the tissue. This applies for the rest of your ms, when you mention habitat use or habitat diversity. For this reason, I recommended in my previous comments to keep the terminology as feeding resources and remove the habitat/microhabitats.

96 connect this sentence with the following paragraph

100 morphological characteristics

101 in addition, ditches are probably more contaminated…

103 protected by a riparian buffer?

104 act as a source of pesticides? How?

109 I personally find the word comparable vague because I don’t know if you mean they can be compared with one being larger than the other (which, of course, they can, using isotopic compositions!) or they are similar.

Methods

150 necessary for the isotopic analyses. Moreover, how did you estimated the weight in the field?

148-149 however, in your response to my previous comment, you said that taxa were actually sorted more than at the family level (i.e., you did not gather individuals from different genera in the same group).

that could explain

225 by lack of information or because the family includes species from different feeding groups?

move this paragraph up to line 258
what is the difference from TA from the previous paragraph? Were the metrics in the previous paragraph not calculated as well for FFG in ponds and ditches?
meaning that in practical terms, TA, SEA.B and SEA.c provide the same information: niche width. Any reason for including the three of them?
I do not follow this new justification. Both C- and N-corrections were made because one water body showed enrichment of either C or N. Then, why now is only one being considered? Moreover, if the C-correction was not used in the analysis, remove the section where it is explained. However, in your Fig. 4 there is a clear difference in the isotopic space with and without correction, so it does have an effect.
my previous question was not answered. Why EC and O2 have small sample size but temperature and pH don’t? they were all measured with a probe. Additionally, in the text and in your answer, it was stated they were measured once, then why there is variability during the day?
this are the number of samples, correct? Then, I recommend not using N since it might be confusing with nitrogen.

Results

then again, you said that the maize correction was not necessary.
significantly distinct
closing dash is missing (and they should be m-dashes)… are significantly larger…

Why is Fig. 5 again only corrected by N? It is very confusing how the authors are mixing results regarding the corrections. This makes very hard to assess the outcomes of these analyses.

…for primary consumers, a mean value 26.5 N for omnivores, and a mean value of 42.25 N for predators… See my previous comment about the use of N for sample size

Moreover, I assume primary consumers refer to grazers, collectors, and shredders? Please, stick to the same terms throughout the manuscript, in this case, terms of table 2. Otherwise, explain why this grouping was made.

groups
I see how my previous comment was not addressed. Your method does not estimate abundance, because you stopped counting until reaching a certain number of individuals. Therefore, you cannot argue that they were observed in larger numbers. Could taxa group be replaced simply by order?
is M the mean? This is not an standard acronym. Do these values correspond to the column Pesticide detections in Table 4? The column does not have units.

Discussion

but you did find differences in the C range, TA, distance to centroid, MNND (Fig. 5). However, this is because this figure was not corrected by C. The fact that some figures/analyses are corrected and some others are not makes the interpretation difficult. Which results are to be used, with or without correction? And based on that, are or are there not differences?

This applies to the whole discussion since some paragraphs seem to be based on corrected values (e.g., paragraph 2) and others not (paragraph 1).

expected
is higher in ponds
led to enriched values
these are not diversity values. These are the number of samples that you used, which should not be used as a diversity metric. A diversity metric is, for example, the number of families.
but again, this is highly speculative since you didn’t measure primary production. More prey could also come from filterers, gatherers, or shredders consuming FPOM or CPOM.
ok but highest densities are not to be compared with higher diversity. They could have found higher densities of a few (or even one) taxa, while you found higher diversity (more families) or predators.

408-411 but these two sentences are completely contradicting each other!

but nutrient concentration was higher in ditches.

In general, this paragraph is very confusing and contradictory.

it seems to be a typo: more less
I partly disagree. The shift could occur in polluted water bodies, or in any water body for multiple reasons. Mainly, resource availability, which changes seasonally, according to the longitudinal position, riparian structure, etc… that is why they are omnivores.
how do you conclude they are highly abundant?
see my comment in the intro about why isotopic compositions are not meant to be used for habitat use, but for feeding resources, which are not the same. Also, keep consistency between habitat and microhabitat-unless you mean something different-, although my suggestion is to remove the habitat part.
do you mean autumn?

**Do you want your identity to be public for this peer review?** For information about this choice, including consent withdrawal, please see our Privacy Policy

Reviewer #1: **Yes: ** Daniela Cortes

---

## [Author Response · Author response to Decision Letter 3]

18 Aug 2025

Author’s response:

We thank the Academic Editor for handling our manuscript and for selecting qualified reviewers. We greatly appreciate receiving such a constructive and detailed review. In response, we have revised the manuscript accordingly again and believe we have addressed all questions and concerns raised. Below, we provide the reviewer’s comments along with our corresponding responses.

Comments to the Author

1. If the authors have adequately addressed your comments raised in a previous round of review and you feel that this manuscript is now acceptable for publication, you may indicate that here to bypass the “Comments to the Author” section, enter your conflict of interest statement in the “Confidential to Editor” section, and submit your "Accept" recommendation.

Reviewer #1: (No Response)

2. Is the manuscript technically sound, and do the data support the conclusions?

Reviewer #1: Partly

3. Has the statistical analysis been performed appropriately and rigorously?

Reviewer #1: No

4. Have the authors made all data underlying the findings in their manuscript fully available?

Reviewer #1: Yes

5. Is the manuscript presented in an intelligible fashion and written in standard English?

Reviewer #1: Yes

6. Review Comments to the Author

Reviewer #1: I have finished reviewing the manuscript for the third time. The manuscript has improved compared to the previous versions. However, I still have significant comments that, in my opinion, must be addressed before publication. I regret to say, once more, that the authors’ responses are occasionally not satisfactory. Moreover, while I appreciate their responses and explanations directed to me, my request and questions are finally made to improve understanding from the readers. In that sense, replying to me, but not fully addressing the issues in the text is not helpful for future readers of their work. Some explanations and changes made from the previous version to this one turned out to be more confusing to me. Specifically, the maize corrections for C. Some analyses and figures are presented with the correction, which leads to the conclusion that there are not differences between water body types, whereas others are presented without the correction, in which case significant differences emerged. In my previous comments, I asked about the different results when correcting versus not correcting, and the answer was that it was only intended as an additional visualization. However, the conclusions are dramatically different between the two methods.

Additional comments are detailed below.

Author’s response:

We thank the reviewer for the time that she invested in evaluating our manuscript again and we hope that we have addressed all the concerns raised by the reviewer now.

Regarding the mentioned corrections for maize I would like to stress that we did the correction for only one pond to be able to compare ponds and ditches and the influences on them of the agricultural stressors (nutrients and pesticides) in general without looking at the extreme effects occurring only in a single pond. Those extreme effects would override the general effects of the stressors we aimed at. The effects of manure fertilization in ditches were corrected as well to check if we can see differences for the stressor pesticides between the two distinct water body types. I made this explanation also more clear in the text. See lines 298 – 304.

Furthermore for some questions the values were corrected for C and N. This was done to see whether differences among FFGs exist beyond these obvious factors. This explanation was also added in the method section, lines 292 – 295.

Comments reviewer 1 with authors responses:

Abstract

15 delete only, as they are not only based on taxonomy, they also account for abundance.

Addressed

30 again, you didn’t measure abundance because you stopped counting when reaching 10 individuals. They were more diverse because you identified more taxa.

Addressed. “No clear pattern emerged when comparing the FFGs in the two water body types, but some differences were found, for example in FFG predator, which were found in the ditches with several different taxa and were thus more diverse.”

Introduction

54 for measuring biodiversity in ponds and ditches affected by agricultural activities… or something similar that provides context for the sentence.

Addressed as suggested: “Thorough knowledge on the status and indicators for measuring biodiversity in ponds and ditches affected by agricultural activities is needed.”

86 taxonomic group since taxa is plural

Addressed.

87 for resource use or consumption

Addressed: “….proxies for resource use, …”

91 the habitat use is not reflected in isotopic compositions because it is not incorporated in the tissue. This applies for the rest of your ms, when you mention habitat use or habitat diversity. For this reason, I recommended in my previous comments to keep the terminology as feeding resources and remove the habitat/microhabitats.

I removed the term microhabitats to avoid confusion, but I still disagree about the term habitat.

Isotopic compositions give information on the habitat occupied by organisms. See also: Newsome, Seth D.; Del Martinez Rio, Carlos; Bearhop, Stuart; Phillips, Donald L. (2007): A niche for isotopic ecology. In: Frontiers in Ecology and the Environment 5 (8), S. 429–436. DOI: 10.1890/060150.1. “Stable isotope analysis provides quantitative information on both resource (bionomic) and habitat (scenopoetic) use commonly utilized to define ecological niche space”

See also as an example: https://www.sciencedirect.com/science/article/pii/S2053716622000032

96 connect this sentence with the following paragraph

Addressed as follows: “We aimed to assess the influence of agricultural stressors, measured as water toxicity from pesticide residues and eutrophication, on the isotopic niche of FFGs in pond and ditch communities, using a geometric approach originally developed by (24). Given that ponds and ditches are likely affected differently by these stressors, with consequent impacts on habitat availability and quality, we hypothesize that such variations ultimately shape benthic invertebrate food webs and niche size (33,34).

100 morphological characteristics

Addressed.

101 in addition, ditches are probably more contaminated…

Addressed.

103 protected by a riparian buffer?

Yes, addressed.

104 act as a source of pesticides? How?

Explanation and literature added in the text as follows: “Ponds, on the other hand, are better protected by a riparian buffer from pesticide inputs, but can act as a sink (e.g. sediment deposition and retention due to long retention times and closed systems, biological uptake in plants) and source (e.g. remobilisation from sediment due to disturbances, contaminated groundwater connection) for pesticide residues in the sediment (35,36).”

109 I personally find the word comparable vague because I don’t know if you mean they can be compared with one being larger than the other (which, of course, they can, using isotopic compositions!) or they are similar.

I changed the question as follows, not using the word comparable: “Do the observed patterns of isotopic niches of FFGs comparable differ between ponds and ditches?

Methods

150 necessary for the isotopic analyses. Moreover, how did you estimated the weight in the field?

Explanation added in the text: “Necessary sample weights were estimated by visual inspection in the field. The exact weighting in the tin capsules for analysis was carried out by the analysing lab, the Cornell Stabel Isotope Facility.”

148-149 however, in your response to my previous comment, you said that taxa were actually sorted more than at the family level (i.e., you did not gather individuals from different genera in the same group).

The lines 148/149 referred to the sorting on site. Here we only sorted organisms into order or family level. Later in the lab we further determined the organisms either to family level or to genus level. See lines 155-157: “In the laboratory, the benthic macroinvertebrates were further taxonomically determined to family (where the order was assigned on site) or genus level when possible and sorted into the FFGs based on the freshwater ecology database (40).”

179 that could explain

Addressed.

225 by lack of information or because the family includes species from different feeding groups?

Could be for both reasons you mentioned here. Additional some taxa are defined as omnivores. Added in the text: “…for a number of reasons, such as lack of information in the database, species with different feeding groups in the family or due to the assignment of the family to the group omnivore by the database.”

move this paragraph up to line 258

Changed as suggested.

what is the difference from TA from the previous paragraph? Were the metrics in the previous paragraph not calculated as well for FFG in ponds and ditches?

As stated in the text lines 269: “Additionally, the Bayesian standard ellipse areas (SEA.B), corrected standard ellipse areas (SEAc), and convex hull area (TA) based on Bayesian interference were calculated.”

Thus the TA here was calculated based on Bayesian statistics and not based on the frequentist statiscical approach as used by Layman. Which is described above.

meaning that in practical terms, TA, SEA.B and SEA.c provide the same information: niche width. Any reason for including the three of them?

Differences explained in more detail in the text: “TA is the total convex hull area enclosing all data points and thus showing the full data range. SEA.B is the Bayesian version of the standard ellipse area, usually compromising 40 % of data points, providing a robust estimate with credible intervals that account for uncertainty (30). The SEA.c based on SEA.B but is adjusted for small sample sizes, a frequentist measure of the core niche that gives a single point estimate (30).

I do not follow this new justification. Both C- and N-corrections were made because one water body showed enrichment of either C or N. Then, why now is only one being considered? Moreover, if the C-correction was not used in the analysis, remove the section where it is explained. However, in your Fig. 4 there is a clear difference in the isotopic space with and without correction, so it does have an effect.

We made the use of corrections more clear in the text: “The calculation of the Bayesian Layman metrics (Layman.B) was performed for the data set with corrected δ15N values, as this was a general phenomenon observed for all ditches and changes throphic position in comparision to ponds and thus could override effects of for example pesticides Maize cultivation was not a general phenomenon like fertilization in the study area, thus the original δ13C values were kept for the following calculations. In other words, the real C-source (resource use) should not be artificially changed because of the niche width, but the N-source had to be scaled to the same level for the two water body types for comparability of trophic position without possible overriding effects of fertilization.”

Yes, of course. The corrections do have an effect this is what figure 4 illustrates. Corrected for N and C values makes the two water body types overlay. The explanation above and added in the text should make clear why δ14N correction is used in the following analysis and C correction is only illustrated in figure 4.

my previous question was not answered. Why EC and O2 have small sample size but temperature and pH don’t? they were all measured with a probe. Additionally, in the text and in your answer, it was stated they were measured once, then why there is variability during the day?

Yes, true. I meant the natural variability during the day, not measured variability. And the small sample size is meant as the small sample size of seven water bodies in general, thus I cannot use a large number of explanatory variables in the dbRDA.

Changed in the text as follows: “Selected environmental variables by the bioenv function, excluding the variables EC and O2, due to small sample size of seven water bodies and the natural expected variability of the variables during the course of the day,…”

Also pH and temperature vary naturally during the day and we only have a small sample size as you said. But those variables were already excluded by the use of the bioenv function, explanined in the lines above. This is the reason they are not mentioned here.

this are the number of samples, correct? Then, I recommend not using N since it might be confusing with nitrogen.

Yes, it is the number of samples. Addressed.

Results

then again, you said that the maize correction was not necessary.

Yes, it was not used for the further analysis as reasons are also clarified above now, but it is still a result worth showing in my opinion. So it is mentioned here as I also show the figure 4.

significantly distinct

Addressed.

closing dash is missing (and they should be m-dashes)… are significantly larger…

Changed as follows: “). The posterior distribution of TA and thus total niche width and the spacing of individuals (mean distance to centroid, CD) are significantly larger in pond communities compared to communities in ditches (Fig. 5 b, e).”

Why is Fig. 5 again only corrected by N? It is very confusing how the authors are mixing results regarding the corrections. This makes very hard to assess the outcomes of these analyses.

I hope the clarifications provided now above in the method section make this clear. Correction for the fertilization in ditches is kept for the analysis as it is a general phenomenon in all ditches and could override other effects. Corrections for the one maize pond is not incorporated in the analysis as it is not a general phenomenon. This correction is only shown in the figure four to illustrate the effect of maize on one pond. This is explained in the manuscript in the methods section in detail.

…for primary consumers, a mean value 26.5 N for omnivores, and a mean value of 42.25 N for predators… See my previous comment about the use of N for sample size

Addressed. N, changed to “isotope samples”

Moreover, I assume primary consumers refer to grazers, collectors, and shredders? Please, stick to the same terms throughout the manuscript, in this case, terms of table 2. Otherwise, explain why this grouping was made.

Addressed. Term defined in brackets in the text. “…(collector/filterer, collector/gatherer, grazer/scraper and shredder)”.

groups

Addressed.

344 I see how my previous comment was not addre

---

## [Decision Letter · Decision Letter 3]

2 Sep 2025

Dear Dr. Trau,

Thank you for submitting your manuscript to PLOS ONE. After careful consideration, we feel that it has merit but does not fully meet PLOS ONE’s publication criteria as it currently stands. Therefore, we invite you to submit a revised version of the manuscript that addresses the points raised during the review process.

**Dear authors, the reviewer has completed the review of your manuscript once again and acknowledged its substantial improvement. However, two major points related to data analysis and discussion are still to be addressed before your manuscript can be accepted for publication. I invite you to take this final step and consider the suggestions thoroughly.**

We look forward to receiving your revised manuscript.

Kind regards,

Marina Vilenica

Academic Editor

PLOS ONE

**Journal Requirements:**

Reviewers' comments:

Reviewer's Responses to Questions

**Comments to the Author**

Reviewer #1: (No Response)

2. Is the manuscript technically sound, and do the data support the conclusions?

Reviewer #1: Yes

3. Has the statistical analysis been performed appropriately and rigorously?

Reviewer #1: Yes

4. Have the authors made all data underlying the findings in their manuscript fully available?

Reviewer #1: Yes

5. Is the manuscript presented in an intelligible fashion and written in standard English?

Reviewer #1: Yes

**Reviewer #1: ** I have finalized revising the manuscript again. I acknowledge the effort of the authors to address all previous comments, which is reflected in the improvement of the manuscript. I still have one general comment and two major issues. My general comment is about the effects of the N and C corrections. I think that the authors could emphasize the reasoning of the corrections in the discussion, to clarify and remind readers of why some figures and results are based on one/both corrections and others are not.

My major issues are:

1. The assignment of taxonomic groups without FFG information to omnivores. This is not trivial, as you are arbitrary assigning NAs to one functional group, which could have an important effect on your results. For example, this could explain the clusters you observed in your analysis. Dealing with NA is an extensively discussed topic in Ecology, and among the multiple approaches to address it, arbitrary assigning values is not a recommended one. My suggestions are as follow: you can remove these taxa from your FFG analysis, and justify this because of the lack of information. You can extend your literature sources to find the appropriate FFG category for such taxa; given the generally coarse taxonomic level you managed (order, family, or genus), I am convinced that this information is available for European taxa. Finally, you can repeat your analysis excluding these taxa and compare to your current results to discard (or not) the influence of your approach.

2. Your discussion about the predator diversity explained by increased primary production. Once more, I consider that adding “However we have to be careful with this conclusions as we did not measure primary

production.” is not enough to address the comment. As I explained before: first, this is especulative as you didn't measure PP; however, you did measure FFG, and you can quantify the number of potential preys available and easily check/discuss if they are more preys in ditches than in ponds. Second, more importantly, one would expect that increasing PP—although only hypothetical—leads to a higher number of primary consumers, particularly grazers. But this is not what you observed. An alternative, which is not discussed in your text, is that predators are not being sustained by a green food web, but by a brown one. This is, instead of PP, there are more detritus sustaining preys (but this also needs to be supported by your own results) and, therefore, sustaining predators. As I mentioned in my previous comment, more available prey could come from filterers, gatherers, or shredders consuming FPOM or CPOM.

**Do you want your identity to be public for this peer review?** For information about this choice, including consent withdrawal, please see our Privacy Policy

Reviewer #1: **Yes: ** Daniela Cortes

---

## [Author Response · Author response to Decision Letter 4]

15 Oct 2025

Author’s response:

We thank the Academic Editor for handling our manuscript and for selecting qualified reviewers. We greatly appreciate receiving such a constructive and detailed review. In response, we have revised the manuscript accordingly again and believe we have addressed all questions and concerns raised. Below, we provide the reviewer’s comments along with our corresponding responses.

Comments to the Author

1. If the authors have adequately addressed your comments raised in a previous round of review and you feel that this manuscript is now acceptable for publication, you may indicate that here to bypass the “Comments to the Author” section, enter your conflict of interest statement in the “Confidential to Editor” section, and submit your "Accept" recommendation.

Reviewer #1: (No Response)

2. Is the manuscript technically sound, and do the data support the conclusions?

Reviewer #1: Yes

3. Has the statistical analysis been performed appropriately and rigorously?

Reviewer #1: Yes

4. Have the authors made all data underlying the findings in their manuscript fully available?

Reviewer #1: Yes

5. Is the manuscript presented in an intelligible fashion and written in standard English?

Reviewer #1: Yes

6. Review Comments to the Author

Reviewer #1: I have finalized revising the manuscript again. I acknowledge the effort of the authors to address all previous comments, which is reflected in the improvement of the manuscript. I still have one general comment and two major issues. My general comment is about the effects of the N and C corrections. I think that the authors could emphasize the reasoning of the corrections in the discussion, to clarify and remind readers of why some figures and results are based on one/both corrections and others are not.

Author’s response:

We thank the reviewer for the time that she invested in evaluating our manuscript again all concerns should be addressed now.

Regarding the general comment on the N and C corrections we added a paragraph in the beginning of the discussion chapter and think that this clarifies the use of the corrections for our readers. The paraghraph was added as follows: “The corrections for the influences of fertilizers (δ15N) and maize (δ13C) were applied to eradicate the overlaying effects, allowing us to set the focus on the differences between either the two water body types or the FFGs in the two water body types and their niche overlaps. In order to investigate whether differences among FFGs niches beyond the obvious factors (fertilization, maize) can be detected we corrected for both isotopes. For comparisions of the two water body types only corrections of δ15N (fertilizer) were applied.”

Comments reviewer 1 with authors responses:

My major issues are:

1. The assignment of taxonomic groups without FFG information to omnivores. This is not trivial, as you are arbitrary assigning NAs to one functional group, which could have an important effect on your results. For example, this could explain the clusters you observed in your analysis. Dealing with NA is an extensively discussed topic in Ecology, and among the multiple approaches to address it, arbitrary assigning values is not a recommended one. My suggestions are as follow: you can remove these taxa from your FFG analysis, and justify this because of the lack of information. You can extend your literature sources to find the appropriate FFG category for such taxa; given the generally coarse taxonomic level you managed (order, family, or genus), I am convinced that this information is available for European taxa. Finally, you can repeat your analysis excluding these taxa and compare to your current results to discard (or not) the influence of your approach.

The table below is a subset from the appendix with all individuals defined as omnivores throughout the study. The added notes indicate the information on feeding preferences we based the decision to define the taxa as omnivores. Notes show that the decision to define the taxa as omnivores is based on information from the literature.

Tables: Taxa assigned to the FFG omnivore with the given information assigned from the freshwater ecology database (Schmidt-Kloiber, Astrid; Hering, Daniel (2015). Ten categories, numbers in brackets in notes) are defined according to Moog (1995).

Note: Please see document attached for the table. The formatting is not shown here, unfortunately.

FFG ponds

taxagroup family_

subfamily genus N notes

omnivore Coleoptera Haliplidae Haliplus sp. 4 assigned to grazers/scrapers (2), miners (4), shredder (1) and predator (3) by freshwater ecology database, ten point system � therefore listed as omnivore

omnivore Diptera Chironomidae NA 33 assigned to grazers/scrapers (2), miners (1), gatherers/collectors (3), active filter feeder (2), predator (3) and parasite (1) by freshwater ecology database� therefore listed as omnivore

omnivore Heteroptera Corixidae Sigara sp. 9 knowledge is contradictory, assigned to gatherers/collectors (10) by freshwater ecology database, feeding habitas according to Tachet et al. (Schmidt-Kloiber and Hering 2015) are more diverse as well as according to Hädicke et al. 2017, so we also assigned them as omnivores

omnivore Heteroptera Corixidae NA 22 knowledge is contradictory, assigned to grazers/scrapers (2), miners (3), gatherers/collectors (5) by freshwater ecology database, but from our data and literature (http://publication.nhmus.hu/pdf/folentom/FoliaEntHung_1978_Vol_31_2_19.pdf and Hädicke et al. 2017) they are known to be also predatory or omnivore � therefore listed as omnivores

FFG ditches

taxagroup family_

subfamily genus N notes

omnivore Coleop-tera Haliplidae Haliplus sp. 34 assigned to grazers/scrapers (2), miners (4), shredder (1) and predator (3) by freshwater ecology database � therefore listed as omnivore

omnivore Diptera Chironomidae NA 36 assigned to grazers/scrapers (2), shredder (3), gatherers/collectors (3) and active filter feeder (1) by freshwater ecology database � therefore listed as omnivore

omnivore Diptera Stratiomyidae NA 2 assigned to grazers/scrapers (3), miners (1), gatherers/collectors (3), active filter feeder (2), predator (3) and parasite (1) by freshwater ecology database � therefore listed as omnivore

omnivore Heterop-tera Corixidae NA 34 see Corixidae above

Based on the information given in the tables we did not follow the recommendation to repeat the analysis excluding taxa without information, as we do have information on the taxa. The FFG omnivore is not assigned based on a lack of knowledge. This was written misleading in our previous comment for the reviewer. We apologize for the confusion.

2. Your discussion about the predator diversity explained by increased primary production. Once more, I consider that adding “However we have to be careful with this conclusions as we did not measure primary production.” is not enough to address the comment. As I explained before: first, this is especulative as you didn't measure PP; however, you did measure FFG, and you can quantify the number of potential preys available and easily check/discuss if they are more preys in ditches than in ponds. Second, more importantly, one would expect that increasing PP—although only hypothetical—leads to a higher number of primary consumers, particularly grazers. But this is not what you observed. An alternative, which is not discussed in your text, is that predators are not being sustained by a green food web, but by a brown one. This is, instead of PP, there are more detritus sustaining preys (but this also needs to be supported by your own results) and, therefore, sustaining predators. As I mentioned in my previous comment, more available prey could come from filterers, gatherers, or shredders consuming FPOM or CPOM.

Yes, we did observe larger numbers of primary consumers in ditches. Also, the numbers of grazers are higher in ditches. This is all listed in table 2. As suggested, we added the potential numbers of available prey in the discussion chapter and added a discussion and literature on the raised point of the alternative of a brown food web sustaining the predators as follows:

“This was underlined by the results, which showed that the potential number of prey was higher in ditches (mean primary consumers = 44.5 N and mean omnivores = 26.5 N), compared to ponds (mean primary consumers = 36 N and mean omnivores = 22.6 N). However, we have to be careful with this conclusion as we did not measure primary production. Alternatively, predators could be sustained by the bown food web to a higher extent in ditches (mean shredder = 61 N) than in ponds (mean shredder = 31 N). This would also be in line with an earlier modelling study indicating that predators may rely on energy derived from the brown food web but also underlines that brown and green food webs are interacting and cascading effects are theoretically in place (60). This could be especially true for non-natural sites, as a study by (61) found that predators rely on detrivores more in restored sites than in reference sites.”

Regarding your second point raised in the comments. We do have higher numbers of primary consumers in ditches, which are most probable characterized by a higher primary production. See text part added above and table two in the manuscript.

Literature:

Hadicke, Christian W.; Redei, David; Kment, Petr (2017): The diversity of feeding habits recorded for water boatmen (Heteroptera: Corixoidea) world-wide with implications for evaluating information on the diet of aquatic insects. In Eur. J. Entomol. 114, pp. 147–159. DOI: 10.14411/eje.2017.020.

Moog, O. (ed.) (1995): Fauna Aquatica Austriaca - A Comprehensive Species Inventory of Austrian Aquatic Organisms with Ecological Notes. Federal Ministry for Agriculture and Forestry, Wasserwirtschaftskataster Vienna: loose-leaf binder.

Schmidt-Kloiber, Astrid; Hering, Daniel (2015): www.freshwaterecology.info – An online tool that unifies, standardises and codifies more than 20,000 European freshwater organisms and their ecological preferences. In Ecological Indicators 53, pp. 271–282. DOI: 10.1016/j.ecolind.2015.02.007.

7. PLOS authors have the option to publish the peer review history of their article (what does this mean?). If published, this will include your full peer review and any attached files.

Do you want your identity to be public for this peer review? For information about this choice, including consent withdrawal, please see our Privacy Policy.

Reviewer #1: Yes: Daniela Cortes

---

## [Decision Letter · Decision Letter 4]

23 Oct 2025

Dear Dr. Trau,

Thank you for submitting your manuscript to PLOS ONE. After careful consideration, we feel that it has merit but does not fully meet PLOS ONE’s publication criteria as it currently stands. Therefore, we invite you to submit a revised version of the manuscript that addresses the points raised during the review process.

**ACADEMIC EDITOR:** Dear authors, many thanks for thorough revision, the Reviewer has no additional requests. However, I have made several minor comments and suggestions in the word document of your manuscript that need to be considered before your manuscript can be accepted for publication.

We look forward to receiving your revised manuscript.

Kind regards,

Marina Vilenica

Academic Editor

PLOS ONE

Journal Requirements:

Reviewers' comments:

Reviewer's Responses to Questions

**Comments to the Author**

Reviewer #1: All comments have been addressed

2. Is the manuscript technically sound, and do the data support the conclusions?

Reviewer #1: Yes

3. Has the statistical analysis been performed appropriately and rigorously?

Reviewer #1: Yes

4. Have the authors made all data underlying the findings in their manuscript fully available?

Reviewer #1: Yes

5. Is the manuscript presented in an intelligible fashion and written in standard English?

Reviewer #1: Yes

Reviewer #1: (No Response)

**Do you want your identity to be public for this peer review?** For information about this choice, including consent withdrawal, please see our Privacy Policy

Reviewer #1: **Yes: ** Daniela Cortes Guzman

---

## [Author Response · Author response to Decision Letter 5]

24 Oct 2025

Author’s response:

We thank the Academic Editor for handling our manuscript and for selecting qualified reviewers. We greatly appreciate that all the reviewers comments were handled successfully and there are no additional request. We have handleded the comments made by the editor in the manuscript. Below, we provide the editors’s comments along with our corresponding responses.

Comments to the Author

1. If the authors have adequately addressed your comments raised in a previous round of review and you feel that this manuscript is now acceptable for publication, you may indicate that here to bypass the “Comments to the Author” section, enter your conflict of interest statement in the “Confidential to Editor” section, and submit your "Accept" recommendation.

Reviewer #1: All comments have been addressed

2. Is the manuscript technically sound, and do the data support the conclusions?

Reviewer #1: Yes

3. Has the statistical analysis been performed appropriately and rigorously?

Reviewer #1: Yes

4. Have the authors made all data underlying the findings in their manuscript fully available?

Reviewer #1: Yes_______________________________________

5. Is the manuscript presented in an intelligible fashion and written in standard English?PLOS ONE does not copyedit accepted manuscripts, so the language in submitted articles must be clear, correct, and unambiguous. Any typographical or grammatical errors should be corrected at revision, so please note any specific errors here.

Reviewer #1: Yes

6. Review Comments to the Author

Reviewer #1: (No Response)

7. PLOS authors have the option to publish the peer review history of their article (what does this mean?). If published, this will include your full peer review and any attached files.

Do you want your identity to be public for this peer review? For information about this choice, including consent withdrawal, please see our Privacy Policy.

Reviewer #1: Yes: Daniela Cortes Guzman

Comments by the editor: Dear authors, many thanks for thorough revision, the Reviewer has no additional requests. However, I have made several minor comments and suggestions in the word document of your manuscript that need to be considered before your manuscript can be accepted for publication.

Author’s response:

We thank the editor for the time that she invested in evaluating our manuscript all concerns should be addressed now. See below for detailed explanations. Comments by the editor or lines given in black and authors responses in green.

Line 10: Addressed as suggested.

Line 39: Changed to: “Agricultural land covers…”

Line 59: Addressed as suggested.

Line 95: Addressed as suggested.

Line 130: Why didnt you choose the same number of sites per habitat type?

Unfortunately, there were only three ponds in the project study region that had enough water for sampling.

Line 136: Sometimes your sites are refer to as sampling locations, sometimes sampling sites, sometimes waterbodies - please choose only one term and use it in the whole manuscript.

Adressed. Changed to sites throughout the manuscript.

Line 149: Microhabitats/different substrate types

The term microhabitat was deleted throughout the review process. In the methods sampled areas within the water bodies are also described as habitats. Also, the sampled habitats include reed or macrophytes so we are not just sampling different substrate types. Thus, I would like to keep the term habitat.

Lines 152-157: How many samples (individuals) you had per habitat (ponds/lakes) in total?

This information was not recorded. Samples from the habitats were pooled. Sampling was carried out in each habitat seperately first to make sure that all present invertebrate groups were sampled. There is an overview of the samples for pond/ditch in table 2.

Line 156: Addressed as suggested.

Line 163: Addressed as suggested.

Line 188: Addressed as suggested.

Line 247: Addressed as suggested.

Line 252: Addressed as suggested.

Line 292: Addressed as suggested.

Line 293: Addressed as suggested.

Line 312: Addressed as suggested.

Lines 321 – 323: Addressed as suggested.

Line 355: Functional groups are sometimes italic, sometimes not.

Functional groups are written in italics throughout the manuscript.

Line 373: It is better to use the full terms in the legends of the figures and tables.

Addressed as suggested.

Line 379: Water parameters were only measured once, and in one spot?

Yes, water parameters were measured once per sampling site. They are measured in-situ in the mixed water sample taken. I made this more clear in the methods lines 187 - 189 as follows: „Environmental water parameters [oxygen content, electrical conductivity (EC), pH, and temperature] were measured in-situ once during sampling in the mixed water sample using a portable meter (WTW Multi 3430 IDS with OxiCal-SL FDO 925, TetraCon 925 and SenTix 940 probes, Germany).

Table 3: Add in the legend with water temperature explanation.

Abbrevation “Temp.” added in table description.

Table 3: Please use the same number of decimal places within a particular parameter, e.g. in pH, all need to have one decimal place, NH4 two etc.

Addressed as suggested.

Line 385: It should be avoided to start a sentence with a number

Addresses as follows: ”The majority of the total pesticides detected (ponds and ditches) are herbicides, accounting for 81%, while 19% are fungicides.”

Line 386: SD shall have the same number of decimal places as the mean.

Addressed as suggested.

Line 392: Addressed as suggested.

Line 406: Addressed as suggested.

Line 426/427: Please use the same number of decimal places for the FFG means, one or two, does not matter, but should be consistent - please correct where necessary

Addressed as suggested.

Line 433: Addressed as suggested.

Line 438: Addressed as suggested.

Line 451: Addressed as suggested.

Line 452: Addressed as suggested.

Line 472: Not changed to microhabitats as suggested. See explanation for keeping the term habitat above.

Line 476: Addressed as suggested.

Line 490: Addressed as suggested.

Line 496: Addressed as suggested.

Line 503: Addressed as suggested.

Line 504/505: Reference?

Application follows the regulations for the pesticide product and the time of crop development. Changed in the text as follws and references added: “Pesticide residues in autumn and winter are lower, as pesticides are mainly applied in spring and early summer due to the general agricultural practice to establish crops such as oilseed rape, maize or cereals during the main growing season (69,70).”

Line 525: Addressed as suggested.

---

## [Editor Report · Decision Letter 5]

28 Oct 2025

Assessing agricultural effects on benthic invertebrate communities in ponds and ditches using δ15N and δ13C isotope niches

PONE-D-25-03786R5

Dear Mrs. Trau,

We’re pleased to inform you that your manuscript has been judged scientifically suitable for publication and will be formally accepted for publication once it meets all outstanding technical requirements.

Kind regards,

Marina Vilenica

Academic Editor

PLOS ONE

---

## [Editor Report · Acceptance letter]

PONE-D-25-03786R5

PLOS ONE

Dear Dr. Trau,

I'm pleased to inform you that your manuscript has been deemed suitable for publication in PLOS ONE. Congratulations! Your manuscript is now being handed over to our production team.

Kind regards,

on behalf of

Dr. Marina Vilenica

Academic Editor

PLOS ONE